# Lanatoside C Inhibits Proliferation and Induces Apoptosis in Human Prostate Cancer Cells Through the TNF/IL-17 Signaling Pathway

**DOI:** 10.3390/ijms26062558

**Published:** 2025-03-12

**Authors:** Sisi Huang, Dongyan Huang, Yangtao Jin, Congcong Shao, Xin Su, Rongfu Yang, Juan Jiang, Jianhui Wu

**Affiliations:** 1NHC Key Lab of Reproduction Regulation, Shanghai Engineering Research Center of Reproductive Health Drug and Devices, Shanghai Institute for Biomedical and Pharmaceutical Technologies, Pharmacy School, Fudan University, Shanghai 200237, China; sisihuang22@m.fudan.edu.cn (S.H.); hdy043@163.com (D.H.); jyt_sphu@163.com (Y.J.); shaocongcongscc@163.com (C.S.); suxiaoxin1982@163.com (X.S.); yangrongfu82@163.com (R.Y.); jiangjuan1106@163.com (J.J.); 2Department of Pharmacology & Toxicology, Shanghai Institute for Biomedical and Pharmaceutical Technologies, Shanghai 200032, China

**Keywords:** prostate cancer, cardiac glycosides, Lanatoside C, tumor necrosis factor, interleukin-17

## Abstract

Prostate cancer remains a leading cause of cancer-related morbidity and mortality among men globally, with limited therapeutic options for advanced and metastatic disease. The therapeutic potential of natural compounds has attracted increasing attention in cancer treatment. Lanatoside C (Lan C), a cardiac glycoside derived from Digitalis lanata, has demonstrated promising anticancer activity across various cancer types. However, its role and mechanisms in prostate cancer remain underexplored. In this study, evidence shows that Lan C significantly inhibits the proliferation of prostate cancer cells, as demonstrated by reduced cell viability, suppressed colony formation, and G2/M cell cycle arrest. Additionally, Lan C promotes apoptosis and inhibits the migration and invasion of prostate cancer cells. Mechanistically, transcriptomic analysis identified differentially expressed genes, which were further validated at both the mRNA and protein levels. Our findings suggest that Lan C exerts its effects by modulating the TNF/IL-17 signaling pathway, influencing the tumor microenvironment and regulating key processes involved in tumor progression, immune response, and apoptosis.

## 1. Introduction

Prostate cancer is the most common malignancy among men worldwide and a leading cause of cancer-related mortality. It is estimated that over 1.1 million men are diagnosed with prostate cancer each year [1]. Due to its long disease course and multiple stages, clinicians have developed a classification system to conceptualize the disease. Based on the primary tumor status, the presence of distant disease (metastatic vs. non-metastatic), and testosterone levels (castrate vs. non-castrate), prostate cancer is categorized into three clinical stages: localized prostate cancer (LPC), metastatic hormone-sensitive prostate cancer (mHSPC), and metastatic castration-resistant prostate cancer (mCRPC) [2,3,4,5]. Despite advancements in diagnostic and therapeutic methods, the management of advanced or metastatic prostate cancer, particularly mCRPC, remains a significant clinical challenge. Current treatment options, including androgen deprivation therapy (ADT), chemotherapy, and targeted therapies, often result in resistance and disease progression. Therefore, there is an urgent need to explore novel therapeutic agents that can target key pathways involved in tumor growth and progression while minimizing adverse effects.

Natural compounds derived from plants have long been considered a valuable source of therapeutic agents due to their structural diversity and bioactivity, which is challenging to replicate synthetically. Cardiac glycosides (CGs), also known as cardiotonic steroids, are natural secondary compounds widely distributed in plants and amphibians. These compounds share a steroid-like structure with an unsaturated lactone ring and exert their primary pharmacological effects by selectively inhibiting the Na^+^/K^+^-ATPase pump, leading to positive inotropic effects. For many years, CGs have been used to treat heart failure and arrhythmias [6,7]. Notable CGs include digoxin, ouabain, oleandrin, and bufalin [6]. It is noteworthy that the Na^+^/K^+^-ATPase pump not only regulates ion homeostasis but also interacts with various signaling pathways involved in cancer cell proliferation, apoptosis, and metastasis. Consequently, the pharmacological applications of CGs have expanded beyond cardiovascular diseases to include cancer, viral infections, inflammation, neurodegenerative disorders, and autoimmune diseases [8,9,10,11,12]. The anticancer potential of CGs was first reported in 1967, and ten years later, scientists observed tumor reduction and a lower recurrence risk in breast cancer patients treated with digoxin. Since then, the anticancer properties of CGs have attracted great interest [12].

Lanatoside C (Lan C) is a cardiac glycoside approved by the U.S. Food and Drug Administration (FDA) for its cardiotonic effects (Figure 1). It has received attention for its promising anticancer and antiviral properties [13,14]. Recent studies have shown that Lan C exhibits significant anticancer activity in various tumor types, including cholangiocarcinoma, pancreatic cancer, non-small-cell lung cancer, cervical cancer, breast cancer, gastric cancer, and colorectal cancer [15,16,17,18,19,20,21]. The mechanisms of its anticancer effects include the induction of apoptosis, cell cycle arrest, disruption of mitochondrial function, and modulation of inflammatory signaling pathways. These multifaceted actions make Lan C a compelling candidate for further research in cancer therapy. However, the specific role and underlying mechanisms of Lan C in prostate cancer remain poorly understood.

The tumor microenvironment (TME) plays a critical role in cancer progression, with inflammation emerging as a marker of malignancy [22]. Pro-inflammatory cytokines, such as tumor necrosis factor-alpha (TNF-α) and interleukin-17 (IL-17), are key components of the inflammatory milieu in prostate cancer. These cytokines activate downstream signaling pathways, including nuclear factor-kappa B (NF-κB) and mitogen-activated protein kinase (MAPK), promoting tumor growth, invasion, and resistance to apoptosis [23]. Notably, TNF-α levels in prostate cancer are associated with disease severity and significantly increase during the metastatic phase. The TNF receptor superfamily regulates two distinct signaling cascades that mediate opposing effects—apoptosis and survival. In prostate cancer, TNF-α-induced signaling promotes cell survival and therapy resistance [24]. IL-17 signaling, closely linked to the progression of various cancers, is particularly relevant in prostate cancer. Of note, Wu et al. found a strong correlation between Cathepsin K (CTSK) and the IL-17 signaling pathway, with the CTSK/IL-17 axis promoting epithelial–mesenchymal transition (EMT) and metastasis in castration-resistant prostate cancer (CRPC) [13,25,26,27]. IL-17 can induce the expression of pro-inflammatory genes alone or synergistically with TNF-α and matrix metalloproteinases [28,29]. Taken together, the TNF/IL-17 signaling axis is involved in creating an immunosuppressive microenvironment that promotes immune evasion by cancer cells. Targeting this pathway represents a promising therapeutic strategy to disrupt the pro-tumorigenic feedback loop and restore immune surveillance.

In addition to its anti-inflammatory properties, Lan C has been reported to induce apoptosis. Studies have shown that Lan C can decrease the expression of Bcl-2 and Bcl-xL, increase Bax expression, and activate caspase-3, leading to apoptosis in cholangiocarcinoma cells [15]. Apoptosis, a form of programmed cell death frequently dysregulated in cancer, plays a crucial role in eliminating damaged or abnormal cells and is tightly regulated by various signaling pathways, including the TNF signaling axis [30]. As previously mentioned, TNF-α is a key cytokine in the inflammatory response, and depending on the cellular context and downstream signaling cascades, it can paradoxically promote both cell survival and death. These effects suggest that Lan C may exert its anti-prostate cancer activity by utilizing the dual roles of TNF-α in immunity and apoptosis.

In this study, we aim to investigate the anticancer effects of Lan C in human prostate cancer cells, with a particular focus on its ability to modulate the TNF/IL-17 signaling pathway and induce apoptosis. Compared to hormone-dependent prostate cancer, hormone-independent cancers are generally more aggressive, less responsive to traditional endocrine therapies, and possess enhanced metastatic capabilities. These tumors often accompany genetic mutations that enable them to adapt and utilize alternative signaling pathways for growth, thus escaping hormonal therapy interventions. Therefore, we selected two hormone-independent prostate cancer cell lines (PC-3 and DU145) for our following experiments. We hypothesize that Lan C exerts its anticancer effects by simultaneously targeting inflammation and apoptosis, thereby disrupting key processes driving tumor progression. To test this hypothesis, we conducted a series of in vitro experiments to evaluate the effects of Lan C on cell proliferation, migration, invasion, and apoptosis. Additionally, transcriptomic analysis was performed to identify differentially expressed genes and affected signaling pathways, followed by validation at both the mRNA and protein levels.

## 2. Results

### 2.1. Effect of Lan C on the Morphology of Prostate Cancer Cells

We observed the effects of different concentrations of Lan C (50 nM, 100 nM, 200 nM, 400 nM) on the growth and morphology of prostate cancer PC-3, DU145, and LNCAP cells after 48 h of treatment in vitro (Figure 2A). As the concentration of Lan C increased, there was a significant inhibition of cell growth in all three prostate cancer cell lines, accompanied by a loss of cell morphology and reduced cell adhesion. At a concentration of 200 nM, the growth of all three prostate cancer cell lines was almost completely suppressed, with significant morphological damage observed. Furthermore, the sensitivity of the three prostate cancer cells to Lan C was different, and the sensitivity was PC-3 > DU145 > LNCAP.

### 2.2. Lan C Inhibits the Growth of Prostate Cancer Cells

The CCK-8 assay was used to assess the effect of Lan C (25–400 nM) on the viability of PC-3, DU145, and LNCAP cells after 24 h, 48 h, and 72 h of treatment, in order to evaluate its growth inhibitory effects on prostate cancer cells. The results showed (Figure 2C–F) that after 48 h and 72 h of treatment, Lan C significantly inhibited the viability of PC-3 cells in all dose groups. After 48 and 72 h of treatment, Lan C with concentrations of 100–400 nM significantly reduced the viability of DU145 cells. Furthermore, 400 nM of Lan C significantly inhibited the viability of LNCAP cells at 24 h, 48 h, and 72 h.

After 24 h of treatment, the half-maximal inhibitory concentrations (IC_50_) for PC-3, DU145, and LNCAP cells were 208.10 nM, 151.30 nM, and 565.50 nM, respectively. After 48 h of treatment, the IC_50_ values for PC-3, DU145, and LNCAP cells were 79.72 nM, 96.62 nM, and 344.80 nM, respectively. After 72 h of treatment, the IC_50_ values for PC-3, DU145, and LNCAP cells were 45.43 nM, 96.43 nM, and 304.60 nM, respectively (Table 1).

### 2.3. Effect of Lan C on the Growth of Normal Prostate Cells

To assess the impact of Lan C on the growth of normal prostate cells, a CCK-8 assay was performed to evaluate the effect of 48 h Lan C treatment on the viability of human normal prostate stromal cells (WPMY-1) and human prostate fibroblasts (HPRFs). The results showed that the IC_50_ of Lan C for HPRF cells was approximately 434 nM, which is significantly higher than the IC_50_ values of 79.72 nM for PC-3 and 96.62 nM for DU145 cells. This indicates that HPRF cells are less sensitive to Lan C compared to the hormone-independent prostate cancer cell lines, DU145 and PC-3 (Figure 2G). Moreover, Lan C has no significant effect on the growth of WPMY-1 (Figure 2G), suggesting that Lan C exhibits a certain degree of selectivity in its action between prostate cancer cells and normal prostate cells.

To further assess the safety range, we calculated the selectivity index of Lan C (Table 2). A selectivity index greater than 1.0 is considered safe and effective, with a higher index indicating a broader safety margin. As observed, all values are greater than 1.0, indicating the safety of Lan C.

### 2.4. Lan C Inhibits Colony Formation in Prostate Cancer Cells

The colony formation assay showed that Lan C significantly inhibited colony formation in both PC-3 and DU145 cells (Figure 2B,H). As the concentration of Lan C increased, colony formation was notably reduced, indicating its strong anti-proliferative effect in both prostate cancer cell lines.

### 2.5. Lan C Promotes Apoptosis in Prostate Cancer Cells

To investigate the effect of Lan C on apoptosis in prostate cancer cells, AnnexinV-FITC/PI dye was used to assess apoptosis (Figure 3A,E). Flow cytometry analysis revealed that, compared to the control group, treatment with 25–400 nM Lan C significantly increased the apoptosis rate in PC-3 cells, with 200 nM and 400 nM Lan C showing a marked induction of apoptosis (Figure 3C). In DU145 cells, treatment with Lan C for 48 h resulted in a significant increase in apoptosis at both 200 nM and 400 nM concentrations (Figure 3G). For clearer images, please refer to Appendix A.

### 2.6. Lan C Induces Cell Cycle Arrest in Prostate Cancer Cells

To further investigate the effect of Lan C on prostate cell proliferation after 48 h, we performed flow cytometry to assess its impact on the cell cycle of PC-3 and DU145 cells (Figure 3B,F). PI staining revealed that, compared to the control group, the high-dose Lan C treatment (400 nM) significantly increased the proportion of PC-3 cells in the G2/M phase (Figure 3D). Additionally, treatment with Lan C resulted in cell cycle arrest in the S phase and G2/M phase in DU145 cells (Figure 3H). For clearer images, please refer to Appendix A.

### 2.7. Lan C Inhibits the Migration Ability of Prostate Cancer Cells

The wound healing assay and the uncoated transwell chamber assay were used to observe the lateral and longitudinal migration abilities of PC-3 and DU145 cells (Figure 4A,C and Figure 5A). More dose group data are shown in Appendix A. Wound healing assays confirmed that, at 48 h, both 100 nM and 200 nM of Lan C significantly inhibited the lateral migration of DU145 cells (Figure 4D). Furthermore, 50–200 nM of Lan C effectively inhibited the migration of PC-3 cells at both 24 h and 48 h (Figure 4B). These results indicate that Lan C inhibits lateral migration of prostate cancer cells, with a stronger inhibitory effect on PC-3 cells. Transwell chamber staining results showed that 50–200 nM of Lan C significantly inhibited the longitudinal migration of both PC-3 and DU145 cells (Figure 5B,C).

### 2.8. Lan C Inhibits Prostate Cancer Cell Invasion

The transwell assay was used to investigate the effect of Lan C on the invasion ability of prostate cancer cells (Figure 5D). More dose group data are shown in Appendix A.Although both PC-3 and DU145 cells exhibit strong invasive ability, Lan C effectively inhibited the invasion of both cell lines, with a more pronounced inhibitory effect observed in DU145 cells (Figure 5E,F).

### 2.9. Transcriptomic Analysis Reveals That Lan C Modulates the TNF/IL-17 Signaling Pathway in PC-3 Cells

To investigate the underlying mechanisms of Lan C’s effects on PC-3 cells, we treated PC-3 cells with or without 200 nM of Lan C for 48 h, followed by RNA extraction for transcriptomic sequencing. Principal component analysis (PCA) of the two groups confirmed that the control and experimental groups represented distinct cell populations, with or without Lan C treatment, respectively (Figure 6A). Subsequently, RNA sequencing (RNA-seq) was employed to identify differentially expressed genes (DEGs) between the control and Lan C-treated groups, with the results visualized as scatter plots, volcano plots, and heatmaps (Figure 6B,C and Figure 7C). Compared to the control group, 200 nM of Lan C treatment led to the identification of 1542 DEGs, with 620 genes upregulated and 922 genes downregulated (Figure 6D). The degree values of all DEGs were ranked, and the top 135 genes were selected to construct a gene–protein interaction (PPI) network (Figure 6E). Gene Ontology (GO) and Kyoto Encyclopedia of Genes and Genomes (KEGG) enrichment analyses were performed on the DEGs, revealing significant enrichment in the TNF, IL-17, and HIF-1 signaling pathways (Figure 7A,B).

### 2.10. Transcriptomic Analysis Reveals That Lan C Modulates the TNF/IL-17 Signaling Pathway in DU145 Cells

To investigate the potential mechanisms underlying the effects of Lan C on DU145 cells, we treated DU145 cells with 200 nM of Lan C or left them untreated for 48 h, followed by transcriptomic analysis of each group. PCA of the two groups confirmed distinct cellular profiles between the control and experimental groups, representing cells treated with or without Lan C (Figure 8A). Subsequently, RNA-seq was used to identify DEGs between the control and Lan C-treated groups, with visualizations provided as scatter plots, volcano plots, and heatmaps (Figure 8B,C and Figure 9C). Compared to the control group, the 200 nM Lan C treatment resulted in 3933 DEGs, including 1913 upregulated and 2020 downregulated genes (Figure 8D). The DEGs were ranked by degree value, and the top 248 genes were used to construct a PPI network (Figure 8E). GO and KEGG enrichment analyses were performed to investigate the functional roles of the DEGs (Figure 9A,B), which were enriched in TNF, P53, and prostate cancer signaling pathways. A Venn diagram comparing the DEGs identified in PC-3 and DU145 cells revealed 514 common DEGs (Figure 10A). A PPI network was constructed for these 514 genes based on their degree values (Figure 10B), which identified several key genes related to the TNF/IL-17 signaling pathway, including *FOS*, *CSF2*, *CXCL1*, *CXCL2*, *CXCL3*, *MMP3*, *MMP1*, *TNFAIP3*, and *BIRC3*.

### 2.11. Regulation of TNF/IL-17 Signaling Pathway Gene Expression by Lan C in PC-3 Cells

To verify the sequencing results, we used real-time quantitative polymerase chain reaction (RT-qPCR) to assess the relative mRNA levels of target genes in the TNF/IL-17 signaling pathway in PC-3 cells. The results showed that Lan C significantly upregulated the expression of *FOS*, *TNFAIP3*, *NFKBIA*, *MMP3*, *CXCL8*, *CCL20*, *MYC*, *PTGS2*, and *IL6*, while downregulating the expression of *MAPK11*, *IL1B*, *MMP1*, and *MAPK13* (Figure 11A–M).

### 2.12. Regulation of TNF/IL-17 Signaling Pathway Gene Expression by Lan C in DU145 Cells

Subsequently, we used RT-qPCR to assess the relative mRNA levels of target genes within the TNF/IL-17 signaling pathway in DU145 cells. The results demonstrated that Lan C significantly upregulated the expression of *FOS*, *TNFAIP3*, *NFKBIA*, *MMP3*, *CXCL8*, *CCL20*, *MYC*, and *PTGS2* in DU145 cells, while it downregulated the expression of *MAPK11* and *IL1B* (Figure 12A–J).

### 2.13. Lan C Upregulates the Expression of FOS and NFKBIA Proteins and Downregulates the Expression of MAPK11, MAPK13, and MMP3 Proteins in PC-3 Cells

To further investigate the molecular mechanisms underlying the effects of Lan C on prostate cancer cells and to validate its regulation of the TNF/IL-17 signaling pathway, immunocytochemical analysis was performed to assess the expression levels of proteins associated with this pathway in PC-3 cells (Figure 13A and Figure 14A). The results demonstrated that in PC-3 cells, the expression of FOS and NFKBIA proteins was significantly upregulated (Figure 13B,C), while the expression of MAPK11, MAPK13, and MMP3 proteins was significantly downregulated (Figure 14B–D). Additionally, Lan C had no significant effect on the expression of TNFAIP3 (Figure 13D).

### 2.14. Lan C Modulates the Expression of FOS, NFKBIA, and TNFAIP3 Proteins and Downregulates MAPK11, MAPK13, and MMP3 Proteins in DU145 Cells

We subsequently evaluated the expression levels of proteins associated with this pathway in DU145 cells using immunocytochemistry (Figure 15A and Figure 16A). The results demonstrated that in DU145 cells, the expression of FOS, NFKBIA, and TNFAIP3 proteins was significantly upregulated (Figure 15B–D), while the expression of MAPK11, MAPK13, and MMP3 proteins was significantly downregulated (Figure 16B–D).

## 3. Discussion

Prostate cancer is one of the most prevalent malignant tumors affecting men and constitutes a significant cause of increased mortality among men globally [31]. Given that androgens play a key role in promoting the progression of prostate cancer, androgen suppression has emerged as an effective strategy to prevent disease progression. Endocrine therapies, such as ADT, including castration therapy and anti-androgen therapies, are commonly employed in the clinical management of prostate cancer [32,33]. However, due to the complexity and heterogeneity of the disease, prostate cancer cells are capable of gradually acquiring the ability to survive under low androgen conditions, ultimately leading to the development of CRPC in the majority of patients, making endocrine treatments ineffective [34]. The use of conventional chemotherapy agents for treating CRPC is associated with low response rates and significant adverse effects, as they not only damage normal cells but also increase the risk of resistance development in tumor cells, compromising the quality of life of patients. Therefore, identifying more optimized therapeutic strategies is a pressing need for both patients and healthcare professionals.

The new use of traditional drugs is a hot research spot in the current medical field. For decades, CGs have been used to treat heart failure and arrhythmias. Interestingly, in the latter half of the 20th century, researchers discovered the anticancer properties of CGs across various cancer types [35], with tumor cells showing higher sensitivity to these compounds than normal cells, which has opened a promising new direction for cancer therapy research. In this study, we investigated the anticancer effects of the cardiac glycoside Lan C on human prostate cancer cells and elucidated its potential underlying mechanisms. Our findings suggest that Lan C inhibits the proliferation and colony formation of prostate cancer cells, suppresses migration and invasion, and induces apoptosis and cell cycle arrest. These effects are at least partially mediated by the modulation of the TNF/IL-17 signaling pathway.

To better understand the dose-dependent effects of Lan C, we selected doses based on the IC_50_ values for PC-3 and DU145 prostate cancer cell lines. The IC_50_ values were used as a benchmark to determine appropriate concentrations for various assays, including colony formation, apoptosis assays, cell cycle analysis, wound healing assays, and transwell migration assays. Doses around the IC_50_ were chosen to reflect concentrations at which Lan C induces significant cellular responses. Additionally, doses above and below the IC_50_ were included to investigate the dose-dependent effects on cancer cell behavior. The lower concentration (below the IC_50_) allowed for the assessment of suboptimal effects, potentially revealing early-stage cellular responses to Lan C. In contrast, the higher concentration (above the IC_50_) enabled the exploration of maximum inhibitory effects and the potential for severe cytotoxicity. This comprehensive dosing strategy provided a thorough evaluation of Lan C’s therapeutic potential across a range of concentrations, offering valuable insights into its efficacy as a treatment for prostate cancer.

TNF can act either as an apoptosis-inducing agent or an inflammatory agent [36]. Uncontrolled cell proliferation is considered as a marker of cancer cells, and the inhibition of proliferation and induction of apoptosis are key indicators of a drug’s effectiveness in targeting tumor cells. Several studies have highlighted the anti-proliferative effects of CG compounds in various cancer cell lines. For example, bufalin induced G2/M phase arrest in human melanoma BRO cells, demonstrating anti-proliferative activity. Jiang et al. reported that bufalin exerted time-dependent anti-proliferative effects on A459 non-small-cell lung cancer (NSCLC) cells, reducing cell viability by enhancing the expression of p53 and p21 (WAF1/CIP1) genes in A549 cells [37]. Additionally, digoxin and its synthetic analog D6-MA inhibit cell cycle progression by downregulating G2/M phase regulators, including cyclin B1, cdc2, and survivin, thus inducing G2/M phase arrest. These compounds also induce apoptosis in NCI-H460 cells through caspase-9 activation via the intrinsic mitochondrial pathway [38,39]. Notably, this anti-proliferative and pro-apoptotic ability is selective. For instance, Anvirzel, a water extract of Nerium oleander tested in clinical trials, demonstrated potent anticancer activity across various cancer types with minimal side effects [40,41,42]. Lan C, a member of the CGs, exhibited anti-proliferative and pro-apoptotic effects in this study, which are consistent with the activity of other CGs in various cancer models. Specifically, following Lan C treatment, the CCK8 assay revealed significant inhibition of cell growth in three prostate cancer cell lines, accompanied by cell decrease and reduced adhesion. Furthermore, Lan C showed limited effects on normal prostate cells, namely WPMY-1 cells and HPRFs. Colony formation assays indicated a significant reduction in the number of colonies formed by PC-3 and DU145 cells, suggesting that Lan C inhibits the clonogenesis ability of prostate cancer cells. Flow cytometry showed that Lan C promotes apoptosis in prostate cancer cells and induces G2/M phase cell cycle arrest. The observed increase in apoptosis treated with the highest concentrations of Lan C is statistically significant, but the overall percentage of apoptotic cells remains relatively low. These data suggest that Lan C may initially induce apoptosis, but further cell death may require longer exposure or higher concentrations to achieve more substantial effects. Additionally, the pro-apoptotic effects of Lan C may be mediated through other cellular mechanisms beyond apoptosis. Beyond its anti-proliferative activity, Lan C also significantly suppressed the migration and invasion potential of prostate cancer cells. Since metastasis is the leading cause of cancer-related mortality in prostate cancer patients, Lan C’s ability to disrupt these processes suggests its potential as a therapeutic agent for metastatic prostate cancer.

It is well known that CGs are involved in the inhibition of the plasma membrane Na^+^, K^+^-ATPase (also known as the Na pump or Na/K pump), leading to increased intracellular Na^+^ and Ca^2+^ levels, and decreased intracellular K^+^ levels [43]. This pump is a transmembrane enzyme that acts as an ion transport protein in the plasma membrane of all mammalian cells. Several Na^+^/K^+^-ATPase-dependent and independent mechanisms may contribute to the anticancer activity of this natural product, including alterations in intracellular levels of Na^+^, K^+^, Ca^2+^, and H^+^; suppression of IL-8 production and the TNF-α/NF-κB pathway; or inhibition of topoisomerase II [28]. Given that the Na^+^/K^+^-ATPase triggers multiple downstream signaling pathways, it suggests that the cellular mechanisms through which CGs exert their effects may extend beyond those reported in the current literature. To explore new mechanisms of action for Lan C in prostate cancer cells, we conducted transcriptome sequencing on Lan C-treated cells. The sequencing results revealed 1542 DEGs in PC-3 cells and 3933 DEGs in DU145 cells, involving a large number of biological processes. Through transcriptomics, particularly KEGG enrichment analysis, and network pharmacology, we identified significant changes in genes related to the TNF/IL-17 signaling pathway, which were subsequently validated at both the mRNA and protein levels. RT-qPCR results showed that treatment with Lan C significantly upregulated the expression of several inflammation- and tumor-related genes, including *FOS*, *TNFAIP3*, *NFKBIA*, *MMP3*, *CXCL8*, *CCL20*, *MYC*, *PTGS2*, and *IL-6*, in PC-3 cells, while downregulating the expression of *MAPK11*, *IL1B*, *MMP1*, and *MAPK13*. In DU145 cells, Lan C upregulated the expression of *FOS*, *TNFAIP3*, *NFKBIA*, *MMP3*, *CXCL8*, *CCL20*, *MYC*, and *PTGS2* and downregulated the expression of *MAPK11* and *IL1B*. Immunocytochemistry results indicated that Lan C upregulated the expression of FOS and NFKBIA proteins and downregulated the expression of MAPK11, MAPK13, and MMP3 proteins in PC-3 cells. In DU145 cells, Lan C upregulated the expression of FOS, NFKBIA, and TNFAIP3 proteins, while downregulating the expression of MAPK11, MAPK13, and MMP3 proteins.

Numerous studies have shown that chronic inflammation is closely associated with the onset and progression of cancer [44,45,46]. Persistent inflammation not only directly contributes to the formation of the tumor microenvironment but also accelerates the development of malignant transformation by promoting processes such as angiogenesis, cell proliferation, tissue invasion, and metastasis [47]. The TNF signaling pathway plays a dual role in immune response and tumorigenesis, as it can both promote tumor initiation and potentially inhibit tumor growth [48]. TNF controls inflammatory signaling and cell survival through binding to two distinct receptors, TNF receptor 1 (TNFR1) and TNF receptor 2 (TNFR2) [49]. TNFR2 is overexpressed in many cancers [50]. This overexpression of TNFR2 can promote the recruitment of tumor cells and activate immune-suppressive cells, thereby aiding the tumor in evading host immune surveillance and advancing tumor progression [51]. Therefore, targeting the TNF/TNFR2 signaling pathway has been considered a promising strategy in cancer immunotherapy [52,53]. Moreover, TNF can also activate the TNFR1 receptor, triggering a series of inflammatory responses and engaging the immune system.

IL-17 is a widely expressed and multifunctional pro-inflammatory cytokine. IL-17 signal transduction begins with the recruitment of Act1 (an E3 ubiquitin ligase) to the cell membrane, which subsequently activates members of the tumor necrosis factor receptor-associated factor (TRAF) family [54]. Among them, activated TRAF6 serves as a critical mediator in signal transduction, triggering several key downstream pathways, including the NF-κB, CCAAT/enhancer binding protein (C/EBP), and MAPK pathways [55,56,57,58,59]. The activation of these pathways ultimately leads to the transcriptional upregulation of several target genes closely associated with inflammatory responses and host defense, including pro-inflammatory cytokines such as *TNF*, *IL-6*, *IL-1β*, *IL-8*, *CXCL1*, *CXCL8*, *CXCL10*, *ICAM1*, and *GM-CSF* and a variety of chemokines such as *CXCL1*, *CXCL2*, *CXCL5*, *CXCL8*, *CXCL10*, *CCL2*, and *CCL20*. Additionally, matrix metalloproteinases (MMPs) associated with extracellular matrix remodeling, such as *MMP1*, *MMP2*, *MMP3*, *MMP9*, and *MMP13*, are also upregulated [58,60].

The TNF/IL-17 signaling pathway plays a crucial role in the inflammatory tumor microenvironment, promoting tumor progression and resistance to apoptosis. Lan C effectively reprograms the tumor microenvironment by downregulating pro-inflammatory cytokines and anti-apoptotic proteins, thereby promoting apoptosis and reducing tumor invasiveness. It is worth mentioning that Lan C’s dual targeting of inflammation and apoptosis provides a unique therapeutic advantage. By modulating key inflammatory mediators on the TNF/IL-17 axis, Lan C disrupts this pro-tumorigenic signaling, offering a novel approach for the treatment of prostate cancer.

Despite the encouraging results, there are some limitations to this study. First, while the in vitro findings provide valuable insights into the anticancer potential of Lan C, further in vivo validation is needed to confirm its therapeutic efficacy and safety. Second, the exact molecular interactions between Lan C and the components of the TNF/IL-17 pathway require deeper investigation. Structural and functional studies will be crucial to fully elucidate these interactions. Finally, the potential synergistic effects of Lan C in combination with existing prostate cancer therapies, such as androgen deprivation therapy or chemotherapy, are worth further exploration.

## 4. Materials and Methods

### 4.1. Chemicals

Lan C (CAS No. 17575-22-3), with a purity of ≥95%, was purchased from Hepai Biotechnology Co., Ltd. (Shanghai, China). The Cell Counting Kit-8 (CCK-8), cell cycle detection kit, DAB horseradish peroxidase color development kit, enhanced immunostaining permeabilization buffer, 4% paraformaldehyde fix solution, QuickBlock™ blocking buffer for immunostaining staining, and QuickBlock™ primary and secondary antibody dilution buffer for immunohistochemistry were all purchased from Beyotime Biotechnology (Shanghai, China). AnnexinV-FITC/PI kits were obtained from Yeasen (Shanghai, China).

### 4.2. Cell Culture

The prostate cancer cell lines PC-3, DU145, and LNCAP were purchased from the National Collection of Authenticated Cell Cultures (NCACC), Chinese Academy of Sciences (Shanghai, China). The human normal prostate stromal immortalized cell line WPMY-1 and human prostate fibroblasts (HPRFs) were obtained from Zhongqiao Xinzhou Biotechnology Co., Ltd. (Shanghai, China). The PC-3 cells were cultured in F12K medium supplemented with 10% fetal bovine serum (FBS), DU145 cells were cultured in MEM medium with 10% FBS, LNCAP cells were maintained in RPMI-1640 medium containing 10% FBS, WPMY-1 cells were cultured in high-glucose DMEM supplemented with 10% FBS, and HPRF cells were grown in FM medium with 2% FBS and 1% FGS. All cells were cultured at 37 °C in a humidified incubator with 5% CO_2_. After preparation, all culture media were supplemented with 0.1% penicillin–streptomycin mix.

The differences between the three tumor cell lines are summarized in Table 3. In this study, after validating the efficacy across all three cell lines, we selected PC-3 and DU145 cells for subsequent experiments, as they are associated with androgen-independent prostate cancer, a more aggressive form of prostate cancer. LNCAP cells are androgen-dependent and exhibit relatively low invasiveness compared to PC-3 and DU145 cells. Moreover, ADT is often an effective treatment for androgen-dependent prostate cancer, leading to a better prognosis. In contrast, PC-3 and DU145 cells are androgen-independent and do not respond to ADT, making them more difficult to treat and representing a clinically challenging phenotype.

### 4.3. Cellular Morphological Observation

Human prostate cancer cell lines PC-3, DU145, and LNCAP were cultured in a humidified incubator at 37 °C with 5% CO_2_ until the cells reached the logarithmic growth phase. The cells were then seeded into 96-well plates at the appropriate density for each cell line (PC-3: 8000 cells/well; DU145: 6000 cells/well; LNCAP: 10,000 cells/well), with 100 μL of cell suspension per well and five replicates per group. After the cells had adhered to the surface, they were divided into two groups: the solvent control group (1‰ DMSO) and the Lan C treatment group (50–400 nM). Following 48 h of treatment, the culture plates were removed, and the morphological changes of the cells were observed and photographed under a microscope.

### 4.4. Cell Viability Assay

Cells in good growth condition, free from contamination, and with a confluence of over 90% were selected for the CCK-8 assay to evaluate the effect of different concentrations of Lan C on cell viability in the five cell lines. Firstly, adherent cells were detached, resuspended, and counted to adjust the cell concentration. The cells were seeded into 96-well plates at the following densities for each time point: for PC-3 cells, 10,000 cells/well for 24 h, 8000 cells/well for 48 h, and 6000 cells/well for 72 h; for DU145 cells, 8000 cells/well for 24 h, 6000 cells/well for 48 h, and 4000 cells/well for 72 h; for LNCAP cells, 12,000 cells/well for 24 h, 10,000 cells/well for 48 h, and 8000 cells/well for 72 h; for WPMY-1 cells, 6000 cells/well for 48 h; and for HPRF cells, 6000 cells/well for 48 h. After cell seeding, 100 μL of cell suspension per well was added to the plates. A series of concentrations of Lan C (25–400 nM) and solvent control (1‰ DMSO) groups were prepared, with five replicates per condition for each cell type. On the following day, the cells were treated with the corresponding Lan C or solvent for 24 h, 48 h, and 72 h. After treatment, 10 μL of CCK-8 solution was added to each well, and the plates were incubated for 2–4 h. Absorbance at 450 nm was measured using a microplate reader (Biotek, Winooski, VT, USA). Cell viability was calculated as follows: cell viability (%) = [(A_sample_ − A_blank_)/(A_vehicle_ − A_blank_)] × 100%. The IC_50_ values for each cell line treated with Lan C were calculated based on the cell viability data, and the selectivity index (SI) was determined. The SI was calculated as SI = IC_50_ of normal cells/IC_50_ of cancer cells.

### 4.5. Colony Formation Assay

After cell digestion, the cell concentration was adjusted to 1 × 10^2^–5 × 10^2^ cells/mL, and 2 mL of the cell suspension was seeded into each well of a 6-well plate. The cells were cultured in a CO_2_ incubator for approximately 24 h until they adhered to the surface. Upon attachment, cells were treated with Lan C at final concentrations of 50 nM, 100 nM, and 200 nM, or with the corresponding volume of solvent (1‰ DMSO), with each group containing three replicates. After 10–12 days of culture, the plates were removed, and 4% paraformaldehyde was added to each well for fixation. After fixation, cells were stained with crystal violet, followed by photographic analysis.

### 4.6. AnnexinV-FITC/PI Double Staining for Apoptosis Detection

Adherent cells from a six-well plate were harvested and suspended. After manual counting, the cells were diluted to achieve a final concentration of 20–25 × 10^4^ cells per well. On the following day, cells were treated with Lan C in final concentrations of 25 nM, 50 nM, 100 nM, 200 nM, and 400 nM, while the control group was treated with an equal volume of solvent (1‰ DMSO). After 48 h of incubation, cells were digested using trypsin without EDTA, followed by centrifugation at 300× *g* for 5 min at 4 °C to collect the cells. After digestion, cells were washed twice with ice-cold phosphate-buffered saline (PBS), with each wash followed by centrifugation at 300× *g* for 5 min at 4 °C. The cells were resuspended in 100 μL of 1× Binding Buffer (Yeasen, Shanghai, China). Subsequently, 5 μL of AnnexinV-FITC and 10 μL of PI Staining Solution were added and gently mixed. The samples were incubated at room temperature, in the dark, for 10–15 min. After incubation, 400 μL of 1× Binding Buffer was added, and the samples were mixed and placed on ice. Flow cytometric analysis was performed by a flow cytometer (BD Biosciences, Franklin Lakes, NJ, USA) within 1 h of preparation.

### 4.7. Cell Cycle Analysis by PI Staining

Adherent cells from six-well plates were harvested and suspended in a medium. After manual cell counting, the suspension was diluted to a concentration of 20–25 × 10^4^ cells per well. The following day, Lan C was added to the cells at final concentrations of 25 nM, 50 nM, 100 nM, 200 nM, and 400 nM. The control group received an equal volume of solvent (1‰ DMSO). After 48 h of incubation, cells were collected by trypsinization, followed by centrifugation at approximately 1000× *g* for 3–5 min to pellet the cells. The supernatant was discarded, and the cell pellet was resuspended in approximately 1 mL of ice-cold PBS. The cell suspension was transferred to a 1.5 mL microcentrifuge tube, and cells were pelleted by centrifugation. The supernatant was carefully removed, and the pellet was resuspended in 1 mL of ice-cold 70% ethanol. Cells were gently mixed and fixed at 4 °C overnight. After centrifugation and removal of the supernatant, cells were resuspended in approximately 1 mL of ice-cold PBS and centrifuged again. The supernatant was discarded, and the cells were resuspended in 500 μL of staining buffer, supplemented with 25 μL of propidium iodide (PI) solution and 10 μL of RNase A. The samples were incubated at 37 °C for 30 min in the dark. The samples were then analyzed using a flow cytometer (BD Biosciences, Franklin Lakes, NJ, USA) within 1 h of preparation. Data were processed and analyzed using FlowJo software v10.8.1 (BD Biosciences, Franklin Lakes, NJ, USA) to determine the distribution of cells across the different phases of the cell cycle.

### 4.8. Cell Migration and Invasion Assays

Before plating, marks were made on the underside of a 6-well plate using a marker and a ruler, with lines spaced 0.5–1 cm apart. Each well was marked with 5–8 lines, each line crossing through the center of the well. Adherent cells were collected from the 6-well plates and resuspended in a medium. After manual counting, the cell suspension was diluted to achieve a final concentration of 20–25 × 10^4^ cells per well. Following plating, the cells were incubated for approximately 24 h in a humidified incubator until most cells had adhered. Once the cells were adherent, a 200 µL pipette tip was used to make linear scratches across the bottom of the well. After scratching, cells were gently washed three times with PBS, and a fresh complete medium containing Lan C with final concentrations of 50 nM, 100 nM, and 200 nM was added. The plates were then incubated in the cell culture incubator, and images were captured at designated time points (0 h, 24 h, and 48 h) under a microscope. The distance of the wound closure was quantified and analyzed using Image J v1.8.0 software.

For transwell migration and invasion assays, transwell inserts with an 8 µm pore polycarbonate membrane were used. Matrigel was applied to the top chamber for the invasion assay, where 10% Matrigel was polymerized on the membrane surface for 24 h at 37 °C (this step was omitted for the migration assay). PC-3 and DU145 cells (5–15 × 10^4^ cells/well for migration, 10–30 × 10^4^ cells/well for invasion) were seeded in 100 µL of serum-free medium containing Lan C at final concentrations of 50 nM, 100 nM, and 200 nM in the upper chambers. A complete medium containing 10% fetal bovine serum was added to the lower chambers. After 48 h of drug treatment, the culture medium was removed, and the cells were washed twice with PBS. Non-migratory cells on the upper surface of the membrane were gently removed with a cotton swab. The cells on the lower surface of the membrane were fixed in 1 mL of 4% paraformaldehyde for 15 min at room temperature and washed twice with PBS. The cells were then stained with 1 mL of crystal violet solution for 20 min and washed three times with PBS. After air-drying, the transwell inserts were transferred to glass slides. Randomly selected fields (6–9 per insert) were observed under a microscope, and the number of cells that migrated through the membrane was quantified using Image J v1.8.0 software.

### 4.9. Transcriptome Sequencing

#### 4.9.1. Total RNA Extraction and Quality Control

Total RNA was extracted from PC-3 and DU145 cells using Trizol reagent (Ambion, Austin, TX, USA), followed by quality assessment using the 2100 Bioanalyzer B.02.09 51725 (Agilent, Santa Clara, CA, USA). RNA samples meeting quality standards were treated with 10 U of DNase I (Takara, Kyoto, Japan) at 37 °C for 30 min to remove any genomic DNA contamination.

#### 4.9.2. mRNA Purification

mRNA was captured using the MagicPure^®^ mRNA Kit (TransGen Biotech, Beijing, China). Briefly, RNA was mixed with 50 μL of magnetic beads and incubated at 65 °C for 5 min, followed by 25 °C for 5 min. The mixture was then placed on a magnetic rack for 5 min to separate the supernatant, which was discarded. Next, 200 μL of WB33II solution was added, the beads were resuspended and mixed, and the mixture was incubated on the magnetic rack for 5 min. The supernatant was discarded. Subsequently, 50 μL of CB33 solution was added, the beads were resuspended and mixed, and the mixture was heated at 80 °C for 2 min. After cooling to 25 °C, 50 μL of BB33 solution was added, and the beads were resuspended and mixed. The mixture was incubated at room temperature for 5 min and placed on the magnetic rack for 5 min, and the supernatant was discarded. Finally, 200 μL of WB33II solution was added, the beads were resuspended and mixed, and the mixture was incubated on the magnetic rack for 5 min. The supernatant was discarded.

#### 4.9.3. cDNA Library Construction

The RNA library was constructed using the TransNGS Stranded RNA-Seq Library Prep Kit for Illumina (TransGen Biotech, Beijing, China). RNA was fragmented by adding RNA Fragmentation Buffer and Primer MIX to the above magnetic beads, followed by incubation at 94 °C for 14 min and rapid cooling on ice. First-strand cDNA synthesis was performed by adding First-Strand Specificity Buffer and First-Strand Enzyme Mix, with the following reaction conditions: 25 °C for 10 min, 42 °C for 15 min, and 85 °C for 10 min. The second-strand cDNA synthesis was then carried out by adding Library Second-Strand Buffer and Library Second-Strand Enzyme Mix, with a reaction at 16 °C for 60 min. After purification using MagicPure Size Selection DNA Beads (TransGen Biotech, Beijing, China), the end repair and A-tailing were performed by adding End-repair and A-tailing Reaction MIX, End-repair and A-tailing Enzyme MIX, and End-repair and A-tailing Enhancer, with the following reaction conditions: 28 °C for 15 min and 68 °C for 15 min. Adaptor ligation was carried out by adding Adapter, Adapter-ligation Buffer, and Adapter-ligation Enzyme Mix, followed by incubation at 25 °C for 15 min. The ligated cDNA was then purified using MagicPure Size Selection DNA Beads. PCR amplification was performed by adding Uracil-DNA Glycosylase, Library Amplification SuperMix II, and Primer, with the following PCR conditions: 50 °C for 5 min, 98 °C for 2 min (pre-denaturation), followed by 15 cycles of 98 °C for 30 s (denaturation), 60 °C for 30 s (annealing), and 72 °C for 30 s (extension). Finally, the library was purified using MagicPure Size Selection DNA Beads, resulting in a sequencing-ready library.

#### 4.9.4. Library Quality Control

Following library construction, three quality control checks were performed to ensure library quality: quantification using Qubit 3.0 fluorometer (Thermo Fisher, Waltham, MA, USA), 2% agarose gel electrophoresis, and high-sensitivity DNA chip analysis.

#### 4.9.5. Sequencing Workflow

Raw data were subjected to quality control, followed by filtering to obtain clean reads. Subsequently, Gene Ontology (GO), Clusters of Orthologous Groups (COG), and pathway analyses were conducted based on the resulting data.

### 4.10. Real-Time Quantitative Polymerase Chain Reaction (RT-qPCR)

Quantitative analysis of messenger RNA (mRNA) expression levels in prostate cancer cells (*n* = 3) was performed using real-time quantitative polymerase chain reaction (RT-qPCR) to compare the control group with Lan C-treated groups (50 nM, 100 nM, 200 nM). Total RNA was efficiently isolated from prostate cancer cells using the RNAeasy™ Kit (R0026, Beyotime Biotechnology, Shanghai, China). RNA purity and concentration were assessed using a spectrophotometer (ACT Gene, Shanghai, China). Reverse transcription of 1 µg of RNA to cDNA was conducted on a PCR machine (Applied Biosystems, Shanghai, China) following the instructions provided with the PrimeScript™ RT Master Mix (Takara, Shanghai, China). Primer sequences were designed using Premier Primer 6 and are detailed in Table 4. mRNA expression levels were quantified using TB Green^®^ Fast qPCR Mix (Takara, Shanghai, China) on a Roche LC480 instrument (Roche, Basel, Switzerland). Relative gene expression analysis was performed using the 2−∆∆Ct method.

### 4.11. Immunocytochemistry

PC-3 and DU145 cells were seeded onto BD Biosciences cell culture slides (354104, Franklin Lakes, NJ, USA) at a density of 8000 cells per well. After overnight incubation to allow for cell attachment, the cells were treated with either a control solution (1‰ DMSO) or varying concentrations of Lan C (50 nM, 100 nM, and 200 nM) for 24 h. The slides were then rinsed three times with phosphate-buffered saline (PBS) to remove residual reagents and subsequently fixed with 4% paraformaldehyde. After fixation, the slides were permeabilized with an immunostaining permeabilization buffer and blocked with QuickBlock™ blocking buffer for 20 min at room temperature to prevent non-specific antibody binding. The cells were incubated overnight at 4 °C with the following primary antibodies: FOS (1:100, AF0255, Beyotime, Shanghai, China), NFKBIA (1:50, AF0255, Beyotime, Shanghai, China), TNFAIP3 (1:50, AF0255, Beyotime, Shanghai, China), MAPK11 (1:50, AF0255, Beyotime, Shanghai, China), MAPK13 (1:50, D162423, Sangon Biotech, Shanghai, China), and MMP3 (1:50, D160611, Sangon Biotech, Shanghai, China). Negative controls were prepared by omitting the primary antibodies during the procedure.

Following incubation with primary antibodies, the cells were exposed to HRP-conjugated goat anti-mouse IgG (1:200, D110087, Sangon Biotech, Shanghai, China) or HRP-conjugated goat anti-rabbit IgG (1:200, D110058, Sangon Biotech, Shanghai, China) for 1 h at room temperature. Afterward, the cells were thoroughly washed three times with PBS to remove unbound antibodies. The staining was developed using a DAB staining solution. Subsequently, the samples were examined under an inverted microscope (AOSVI, Shenzhen, China). For each experimental group, three distinct cell sections were analyzed, and within each section, three random microscopic fields were selected for further evaluation. The obtained cell section images were analyzed using Image-Pro Plus 6.0 software. The integrated optical density (IOD) values, which are proportional to the total amount of the target protein, were quantified. The IOD values were normalized by dividing by the area of the target protein distribution (IOD/area) to obtain the average optical density (AOD), which reflects the protein concentration per unit area. Semi-quantitative expression levels of FOS, NFKBIA, TNFAIP3, MAPK11, MAPK13, and MMP3 were determined using AOD values.

### 4.12. Statistical Analysis

The results are expressed as means ± standard deviation (means ± SD), and statistical analysis was performed using SPSS 26.0 software. For data that followed a normal distribution, one-way analysis of variance (one-way ANOVA) was used for group comparisons. When the data did not meet the assumption of normality, the Kruskal–Wallis H test was applied for group comparisons. For data that satisfied the assumptions of normality and homogeneity of variance, the least significant difference (LSD) method was used for post hoc analysis; for data with unequal variances, Dunnett’s T3 test was employed. *p* < 0.05 was considered statistically significant. Visual representations of the data were generated using GraphPad Prism 9.0 software (GraphPad Software, San Diego, CA, USA).

## 5. Conclusions

Lan C effectively inhibits the proliferation, migration, and invasion of prostate cancer cells, induces apoptosis, and arrests the cell cycle, demonstrating potent anticancer effects in human prostate cancer cells. These effects are mediated through the modulation of the TNF/IL-17 signaling pathway, highlighting its role in reprogramming the inflammatory tumor microenvironment and promoting apoptotic signaling. Our findings suggest that Lan C holds significant potential as a therapeutic agent for prostate cancer, providing a multifaceted approach to targeting tumor progression. Further in vivo studies and clinical validation are required to promote the translational application of Lan C in cancer treatment.

## Figures and Tables

**Figure 1 ijms-26-02558-f001:**
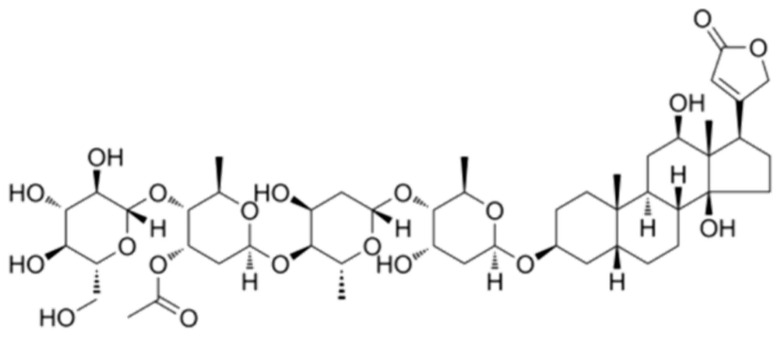
Chemical structure of Lanatoside C.

**Figure 2 ijms-26-02558-f002:**
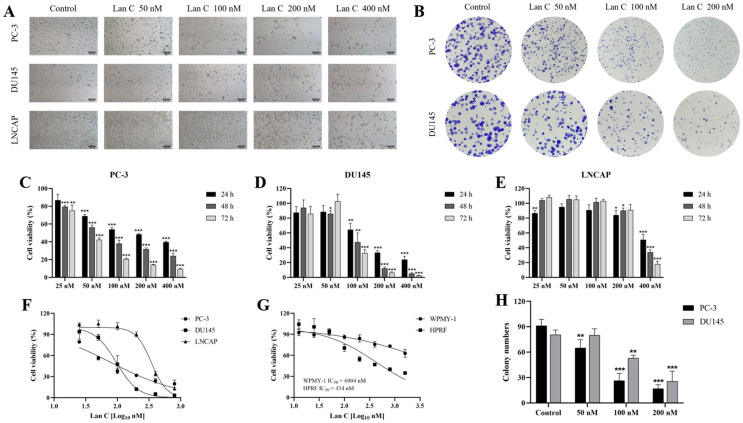
Effect of Lan C on the proliferation of human prostate cancer cells and normal human prostate cells. (**A**) Morphological changes of human prostate cancer cells (PC-3, DU145, and LNCAP) following 48 h of treatment with Lan C (100×, scale bar = 100 μm). (**B**) The effect of Lan C on the colony-forming ability of human prostate cancer cells PC-3 and DU145 (**B**) and statistical histogram (**H**). (**C**–**E**) Effect of Lan C treatment on cell viability in PC-3 (**C**), DU145 (**D**), and LNCAP (**E**) after 24 h, 48 h, and 72 h of treatment. (**F**) Inhibition of proliferation in PC-3, DU145, and LNCAP prostate cancer cells after 48 h of Lan C treatment. (**G**) Effect of Lan C on the proliferation of normal human prostate stromal cell line WPMY-1 and human prostate fibroblasts HPRF after 48 h of treatment. Data are presented as the mean ± standard deviation (SD) of three independent experiments. Statistical analysis was performed after normality testing, using one-way analysis of variance (ANOVA) for between-group comparisons. If homogeneity of variance was assumed, the least significant difference (LSD) test was used; if the variance was heterogeneous, Dunnett’s post hoc test was applied. Comparison with the control group: * *p* < 0.05, ** *p* < 0.01, *** *p* < 0.001.

**Figure 3 ijms-26-02558-f003:**
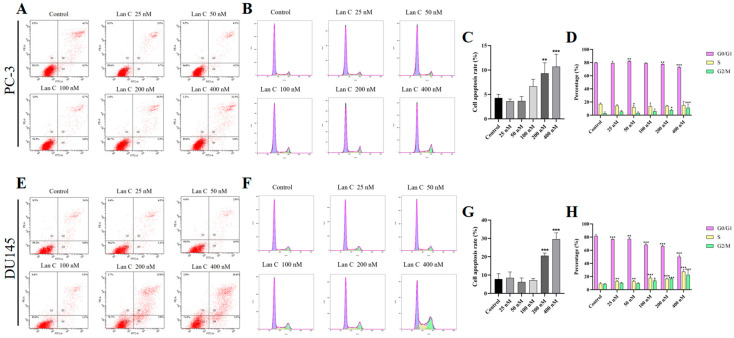
Lan C promotes apoptosis and cell cycle arrest in human prostate cancer cells. Apoptotic scatter plots (**A**) and statistical histogram (**C**) of apoptotic proportions of human prostate cancer cells treated with PC-3 by Lan C for 48 h. Cell cycle distribution diagram (**B**) and statistical analysis diagram (**D**) of human prostate cancer cells treated with PC-3 by Lan C for 48 h. The scatter plot (**E**) of apoptosis and the statistical histogram (**G**) of the proportion of apoptosis in each group after treatment of human prostate cancer cells DU145 by Lan C for 48 h. Cell cycle distribution diagram (**F**) and statistical analysis diagram (**H**) of human prostate cancer cells DU145 treated by Lan C for 48 h. Data are presented as the mean ± SD of three independent experiments. Statistical analysis was performed after normality testing, using ANOVA for between-group comparisons. If homogeneity of variance was assumed, the LSD test was used; if the variance was heterogeneous, Dunnett’s post hoc test was applied. Comparison with the control group: * *p* < 0.05, ** *p* < 0.01, *** *p* < 0.001.

**Figure 4 ijms-26-02558-f004:**
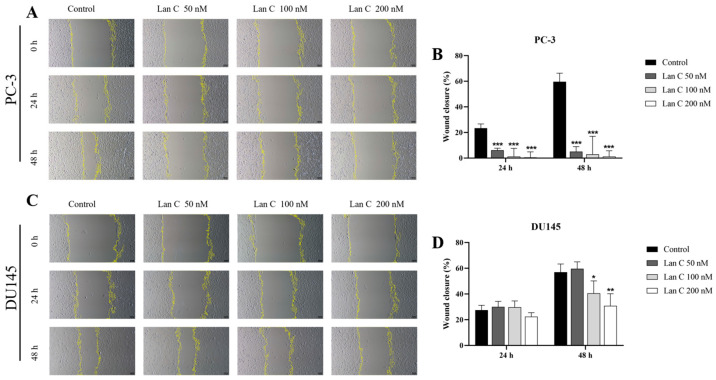
Lan C inhibits lateral migration of human prostate cancer cells. (**A**,**B**) Scratch wound healing assays were performed to assess the migration ability of PC-3 prostate cancer cells after treatment with Lan C for 24 h and 48 h (40×, scale bar = 100 μm). (**C**,**D**) Scratch wound healing assays were performed to assess the migration ability of DU145 prostate cancer cells after treatment with Lan C for 24 h and 48 h (40×, scale bar = 100 μm). Data are presented as the mean ± SD of three independent experiments. Statistical analysis was performed after normality testing, using ANOVA for between-group comparisons. If homogeneity of variance was assumed, the LSD test was used; if the variance was heterogeneous, Dunnett’s post hoc test was applied. Comparison with the control group: * *p* < 0.05, ** *p* < 0.01, *** *p* < 0.001.

**Figure 5 ijms-26-02558-f005:**
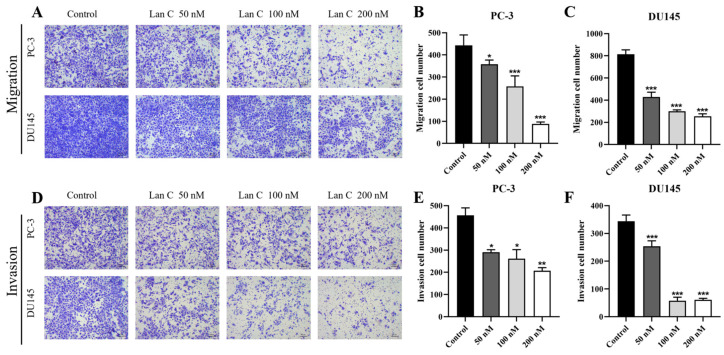
Lan C inhibits the longitudinal migration and invasion of human prostate cancer cells. (**A**–**C**) The effect of Lan C on the longitudinal migration of human prostate cancer cells PC-3 and DU145 after treatment for 48 h (40×, scale bar = 100 μm). (**D**–**F**) The effect of Lan C on the invasion ability of human prostate cancer cells PC-3 and DU145 after treatment for 48 h (40×, scale bar = 100 μm). Data are presented as the mean ± SD of three independent experiments. Statistical analysis was performed after normality testing, using ANOVA for between-group comparisons. If homogeneity of variance was assumed, the LSD test was used; if the variance was heterogeneous, Dunnett’s post hoc test was applied. Comparison with the control group: * *p* < 0.05, ** *p* < 0.01, *** *p* < 0.001.

**Figure 6 ijms-26-02558-f006:**
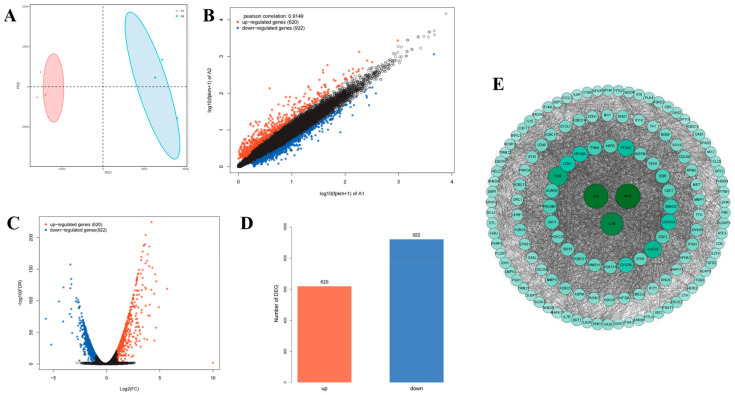
Transcriptomic analysis of the effects of Lan C on human prostate cancer PC-3 cells. (**A**) Principal component analysis (PCA) plots comparing the two groups of PC-3 cells treated with and without Lan C. (**B**) Scatter plot visualizing differentially expressed genes (DEGs) in PC-3 cells treated with 200 nM of Lan C. The screening criteria were |logFC|≥ 1 and FDR ≤ 0.05. Red points represent significantly upregulated genes, blue points represent significantly downregulated genes, and black points indicate genes with no significant changes. (**C**) Volcano plot of DEGs in PC-3 cells treated with 200 nM of Lan C. Red points represent significantly upregulated genes, blue points represent significantly downregulated genes, and black points represent genes with no significant changes. (**D**) The number of differentially expressed genes in PC-3 cells treated with 200 nM of Lan C. (**E**) Protein–protein interaction (PPI) network of DEGs in PC-3 cells treated with 200 nM of Lan C.

**Figure 7 ijms-26-02558-f007:**
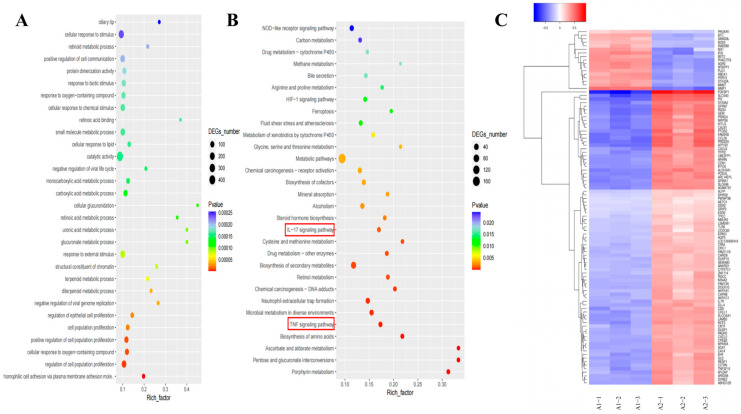
Transcriptomic and network pharmacology analysis of the effects of Lan C on human prostate cancer PC-3 cells. (**A**) Gene Ontology (GO) enrichment analysis of DEGs in PC-3 cells treated with 200 nM of Lan C. The x-axis represents the enrichment factor, which is the ratio of differentially expressed genes enriched in a specific GO term to the total number of background genes sequenced. The y-axis shows the functions associated with the GO terms. The size of the circles indicates the number of differentially expressed genes enriched in each GO term. The color gradient from blue to red represents the unadjusted *p*-value. (**B**) Kyoto Encyclopedia of Genes and Genomes (KEGG) enrichment analysis of DEGs in PC-3 cells treated with 200 nM of Lan C. The x-axis represents the enrichment factor, which is the ratio of differentially expressed genes enriched in a specific KEGG pathway to the total number of background genes sequenced. The y-axis shows the functions associated with the KEGG pathways. The size of the circles indicates the number of differentially expressed genes enriched in each KEGG pathway. The color gradient from blue to red represents the unadjusted *p*-value. (**C**) Heatmap of the expression levels of differentially expressed genes in PC-3 cells treated with 200 nM of Lan C. Each column represents a sample, and each row represents a gene. The color scale represents the gene expression level (log_10_ FPKM), with the color gradient from blue (low expression) to red (high expression).

**Figure 8 ijms-26-02558-f008:**
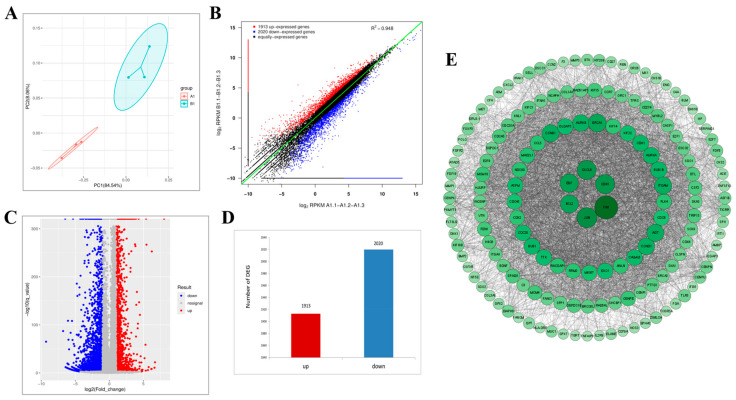
Transcriptomic analysis of Lan C in human prostate cancer DU145 cells. (**A**) PCA of DU145 cells with and without Lan C treatment. (**B**) Scatter plot visualizing DEGs in DU145 cells treated with 200 nM of Lan C. Screening criteria: |logFC| ≥ 1 and FDR ≤ 0.05. Red points represent significantly upregulated genes, blue points represent significantly downregulated genes, and black points represent genes with no significant differential expression. (**C**) Volcano plot of DEGs in DU145 cells treated with 200 nM of Lan C. Red points indicate significantly upregulated genes, blue points represent significantly downregulated genes, and the gray region represents non-significant genes. (**D**) Number of DEGs in DU145 cells after 200 nM of Lan C treatment. (**E**) PPI network of DEGs in DU145 cells treated with 200 nM of Lan C.

**Figure 9 ijms-26-02558-f009:**
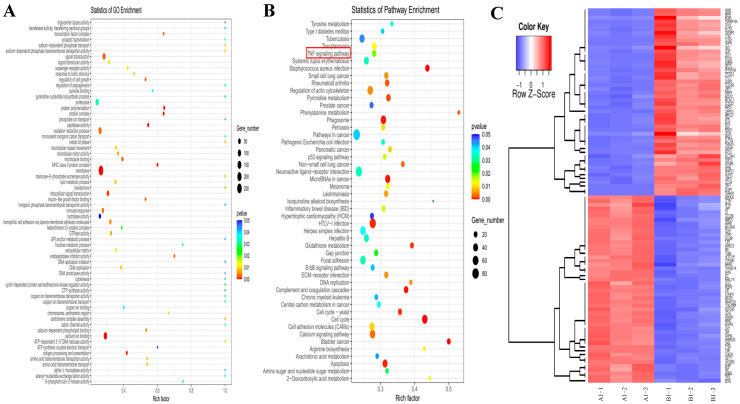
Transcriptomic and network pharmacology analysis of Lan C in human prostate cancer DU145 cells. (**A**) GO enrichment analysis of DEGs in DU145 cells treated with 200 nM of Lan C. The x-axis represents the enrichment factor, which is the ratio of the number of DEGs enriched in a particular GO term to the total number of background genes sequenced. The y-axis denotes the biological functions enriched in each GO term. The circle size reflects the relative number of DEGs enriched in each function. The color gradient from blue to red indicates the unadjusted *p*-value. (**B**) KEGG pathway enrichment analysis of DEGs in DU145 cells treated with 200 nM of Lan C. The x-axis represents the enrichment factor, and the y-axis shows the KEGG pathways enriched with DEGs. Circle size correlates with the number of DEGs enriched in each pathway, with the color gradient from blue to red indicating the unadjusted *p*-value. (**C**) Heatmap of differentially expressed genes in DU145 cells after 200 nM of Lan C treatment. Each column represents a sample, and each row represents a gene. The color scale reflects gene expression levels in the sample (log10 FPKM), ranging from low expression (blue) to high expression (red).

**Figure 10 ijms-26-02558-f010:**
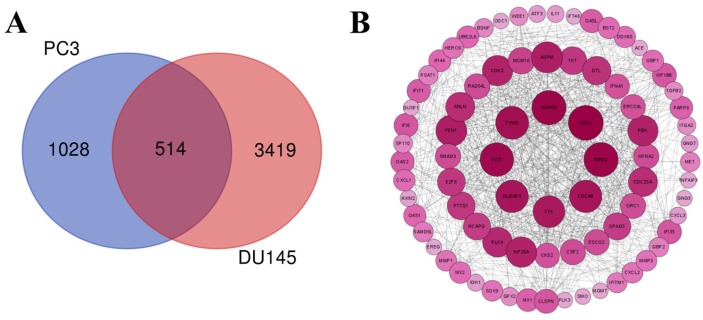
Common DEGs and their interaction network in PC-3 and DU145 cells induced by Lan C treatment. (**A**) Venn diagram of common DEGs between PC-3 and DU145 cells following Lan C treatment. (**B**) PPI network of common DEGs between PC-3 and DU145 cells after Lan C treatment.

**Figure 11 ijms-26-02558-f011:**
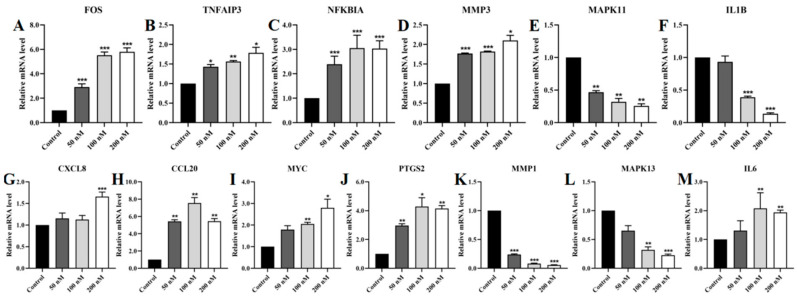
Effect of Lan C on the expression of genes associated with the TNF/IL-17 signaling pathway in PC-3 cells. Lan C significantly upregulates the expression of (**A**) the *FOS* gene, (**B**) the *TNFAIP3* gene, (**C**) the *NFKBIA* gene, (**D**) the *MMP3* gene, (**G**) the *CXCL8* gene, (**H**) the *CCL20* gene, (**I**) the *MYC* gene, (**J**) the *PTGS2* gene, and (**M**) the *IL6* gene in PC-3 cells. Lan C significantly downregulates the expression of (**E**) the *MAPK11* gene, (**F**) the *IL1B* gene, (**K**) the *MMP1* gene, and (**L**) the *MAPK13* gene in PC-3 cells. Data are presented as the mean ± SD of three independent experiments. Statistical analysis was performed after normality testing, using ANOVA for between-group comparisons. If homogeneity of variance was assumed, the LSD test was used; if the variance was heterogeneous, Dunnett’s post hoc test was applied. Comparison with the control group: * *p* < 0.05, ** *p* < 0.01, *** *p* < 0.001.

**Figure 12 ijms-26-02558-f012:**
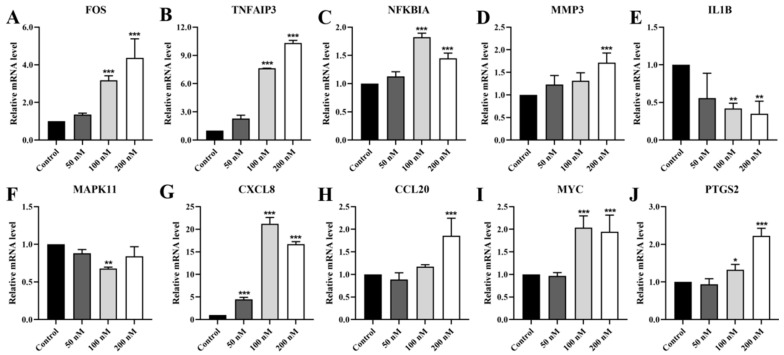
Effect of Lan C on the expression of genes associated with the TNF/IL-17 signaling pathway in DU145 cells. Lan C significantly upregulates the expression of (**A**) the *FOS* gene, (**B**) the *TNFAIP3* gene, (**C**) the *NFKBIA* gene, (**D**) the *MMP3* gene, (**G**) the *CXCL8* gene, (**H**) the *CCL20* gene, (**I**) the *MYC* gene, and (**J**) the *PTGS2* gene in DU145 cells. Lan C significantly downregulates the expression of (**E**) the *IL1B* gene, and (**F**) the *MAPK11* gene in DU145 cells. Data are presented as the mean ± SD of three independent experiments. Statistical analysis was performed after normality testing, using ANOVA for between-group comparisons. If homogeneity of variance was assumed, the LSD test was used; if the variance was heterogeneous, Dunnett’s post hoc test was applied. Comparison with the control group: * *p* < 0.05, ** *p* < 0.01, *** *p* < 0.001.

**Figure 13 ijms-26-02558-f013:**
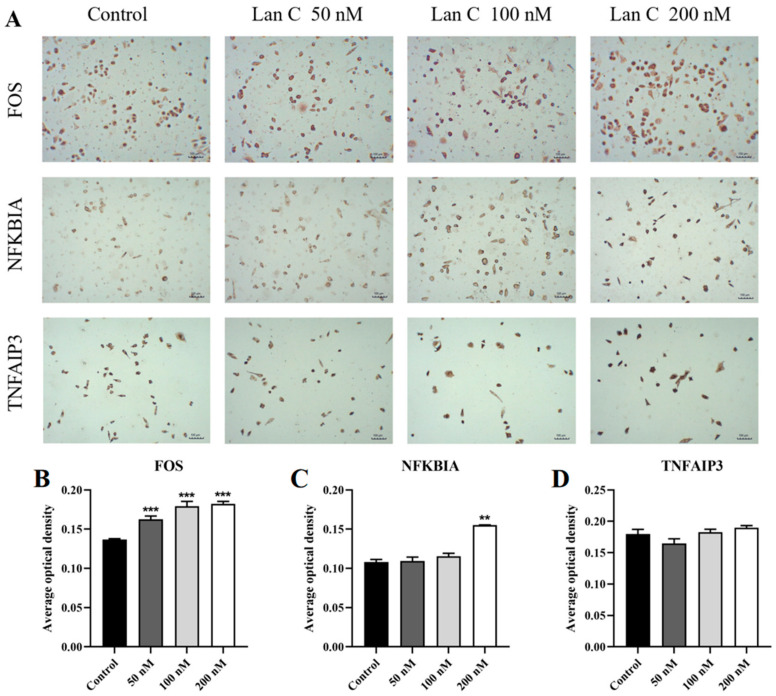
Effect of Lan C on the expression of TNF/IL-17 signaling pathway-related proteins in PC-3 cells. (**A**) Immunocytochemical images of FOS, NFKBIA, and TNFAIP3 in PC-3 cells after treatment with Lan C (100×, scale bar = 100 μm). Lan C significantly upregulates the expression of (**B**) FOS protein and (**C**) NFKBIA protein. (**D**) Effect of Lan C on the expression of TNFAIP3 protein. Data are presented as the mean ± SD of three independent experiments. Statistical analysis was performed after normality testing, using ANOVA for between-group comparisons. If homogeneity of variance was assumed, the LSD test was used; if the variance was heterogeneous, Dunnett’s post hoc test was applied. Comparison with the control group: ** *p* < 0.01, *** *p* < 0.001.

**Figure 14 ijms-26-02558-f014:**
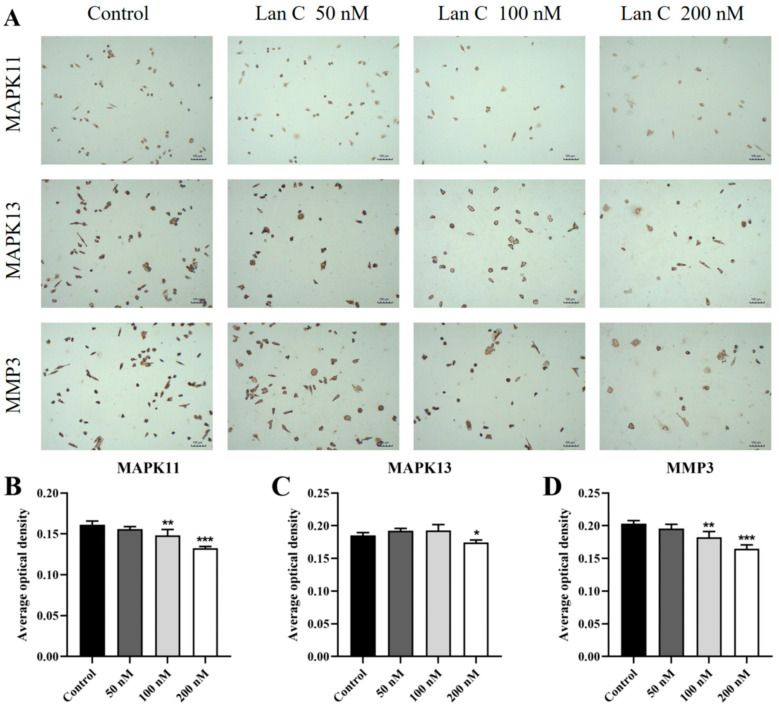
Effect of Lan C on the expression of TNF/IL-17 signaling pathway-related proteins in PC-3 cells. (**A**) Immunocytochemical images of MAPK11, MAPK13, and MMP3 in PC-3 cells after treatment with Lan C (100×, scale bar = 100 μm). Lan C significantly downregulates the expression of (**B**) MAPK11 protein, (**C**) MAPK13 protein, and (**D**) MMP3 protein. Data are presented as the mean ± SD of three independent experiments. Statistical analysis was performed after normality testing, using ANOVA for between-group comparisons. If homogeneity of variance was assumed, the LSD test was used; if the variance was heterogeneous, Dunnett’s post hoc test was applied. Comparison with the control group: * *p* < 0.05, ** *p* < 0.01, *** *p* < 0.001.

**Figure 15 ijms-26-02558-f015:**
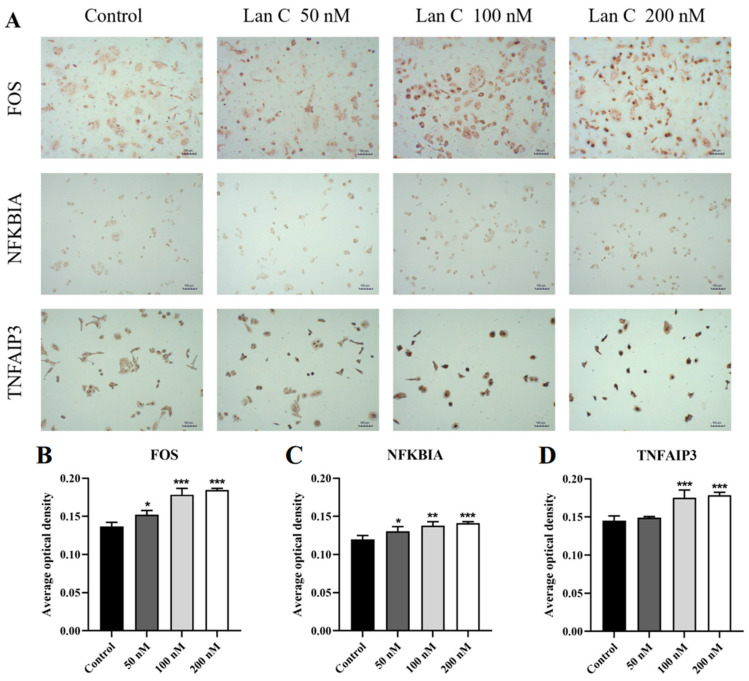
Effects of Lan C on the expression of TNF/IL-17 signaling pathway-related proteins in DU145 cells. (**A**) Immunocytochemical images of FOS, NFKBIA, and TNFAIP3 in DU145 cells treated with Lan C (100×, scale bar = 100 μm). Lan C significantly upregulates the expression of (**B**) FOS protein, (**C**) NFKBIA protein, and (**D**) TNFAIP3 protein. Data are presented as the mean ± SD of three independent experiments. Statistical analysis was performed after normality testing, using ANOVA for between-group comparisons. If homogeneity of variance was assumed, the LSD test was used; if the variance was heterogeneous, Dunnett’s post hoc test was applied. Comparison with the control group: * *p* < 0.05, ** *p* < 0.01, *** *p* < 0.001.

**Figure 16 ijms-26-02558-f016:**
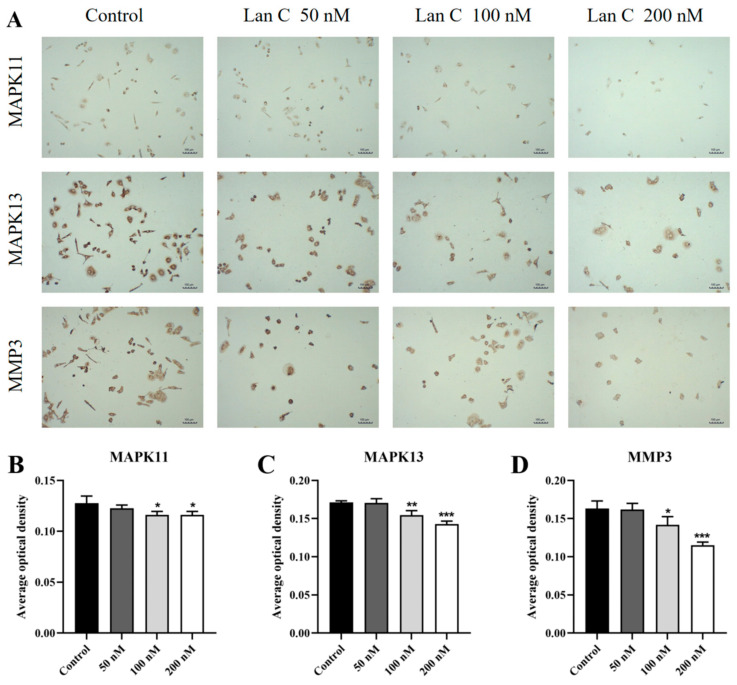
Effects of Lan C on the expression of TNF/IL-17 signaling pathway-related proteins in DU145 cells. (**A**) Immunocytochemical images of MAPK11, MAPK13, and MMP3 in DU145 cells treated with Lan C (100×, scale bar = 100 μm). Lan C significantly downregulates the expression of (**B**) MAPK11 protein, (**C**) MAPK13 protein, and (**D**) MMP3 protein. Data are presented as the mean ± SD of three independent experiments. Statistical analysis was performed after normality testing, using ANOVA for between-group comparisons. If homogeneity of variance was assumed, the LSD test was used; if the variance was heterogeneous, Dunnett’s post hoc test was applied. Comparison with the control group: * *p* < 0.05, ** *p* < 0.01, *** *p* < 0.001.

**Table 1 ijms-26-02558-t001:** IC_50_ values (nM) of Lan C on the inhibition of human prostate cancer cells PC-3, DU145, and LNCAP.

Cell Lines	IC_50_ (24 h)	IC_50_ (48 h)	IC_50_ (72 h)
PC-3	208.10	79.72	45.43
DU145	151.30	96.62	96.43
LNCAP	565.50	344.80	304.60

**Table 2 ijms-26-02558-t002:** Selectivity index values calculated for different cell lines after 48 h of treatment with Lan C.

Cell Lines	IC_50_ (48 h)	Selectivity Index (WPMY-1) ^a^	Selectivity Index (HPRF) ^b^
PC-3	79.72 nM	75.3	5.4
DU145	96.62 nM	62.1	4.5
LNCAP	344.80 nM	17.4	1.3

^a^ Selectivity index (WPMY-1) = IC_50_ of WPMY-1/IC_50_ of cancer cells. ^b^ Selectivity index (HPRF) = IC_50_ of HPRFs/IC_50_ of cancer cells.

**Table 3 ijms-26-02558-t003:** The differences between the prostate cancer cell lines.

Cell Lines	Source	Migration Ability	Response to Androgen
LNCAP	Androgen-dependent prostate cancer	Weak	Androgen-dependent
PC-3	Bone metastasis from androgen-independent prostate cancer	Moderate	Androgen-independent
DU145	Brain metastasis from androgen-independent prostate cancer	Strong	Androgen-independent

**Table 4 ijms-26-02558-t004:** Primer sequences for RT-qPCR.

Genes	Forward Primer (5′-3′)	Reverse Primer (5′-3′)
*GAPDH*	TCCAAAATCAAGTGGGGCGA	AAATGAGCCCCAGCCTTCTC
*MYC*	TTCATAACGCGCTCTCCAAGT	CAGAGCGTGGGATGTTAGTGT
*IL6*	CTTCGGTCCAGTTGCCTTCT	TGGAATCTTCTCCTGGGGGT
*IL1B*	GTTCTTTGAAGCTGATGGCCC	GAAGCCCTTGCTGTAGTGGT
*FOS*	CAAGCGGAGACAGACCAACT	GTGAGCTGCCAGGATGAACT
*CXCL8*	CACTGCGCCAACACAGAAAT	TTCTCAGCCCTCTTCAAAAACTTC
*PTGS2*	GTTCCACCCGCAGTACAGAA	AGGGCTTCAGCATAAAGCGT
*NFKBIA*	CCACTCCACTTGGCTGTGAT	TTCCTCGAAAGTCTCGGAGC
*CCL20*	GCGAATCAGAAGCAAGCAAC	CCGTGTGAAGCCCACAATAAA
*MMP3*	ATCCTACTGTTGCTGTGCGT	GGTTCATGCTGGTGTCCTCA
*TNFAIP3*	TCCACAAAGCCCTCATCGAC	TTCGTTTTCAGCGCCACAAG
*MAPK11*	AGAACGTCATCGGGCTTCT	TGGCACTTGACGATGTTGTT
*MAPK13*	AAGACCTACGTGTCCCCGA	TCGGCTCAGCTTCTTGATGG
*MMP1*	AGAGCAGATGTGGACCATGC	TTGTCCCGATGATCTCCCCT

## Data Availability

The data provided in this study are available upon request to the corresponding author.

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
