# Peer review of "Lanatoside C Inhibits Proliferation and Induces Apoptosis in Human Prostate Cancer Cells Through the TNF/IL-17 Signaling Pathway"

_ijms, 2025, doi:10.3390/ijms26062558_

Round 1

Reviewer 1 Report

Comments and Suggestions for Authors

In this paper the author identify anti-cancer properties of Lanatoside C (Lan C), a cardiac glycoside compound. The authors mainly use two hormone-independent prostate cancer cell lines (PC-3 and DU145), alongwith normal human prostate stromal cell line WPMY-1 and human prostate fibroblasts HPRF, in their experiments. They find Lan C to be anti-proliferative and anti-metastatic by affecting the TNF-alpha signaling pathway. It does not affect the viability of other stromal cells and fibroblasts of the prostrate origin. The findings are very well written but there are a few points that can be improved and corrected. 

1) In figure 2G, the graph for cell viabilty of HPRF can be repeated as it does not very well present the IC50 values which are statistically significant.

2) In the subsequent paper, the figures presenting Protein-protein interaction (PPI) network of DEGs and KEGG pathway analysis are visibly not very clear and I think can be made better.

3)In Figure 11, the conclusions made for NFKBIA and MAPK11 proteins from the immunochemistry data is not very apparent. As a reader, I cannot see the upregulation and downregulation very clearly in DU145 cells. Same issue with MAPK11 protein data in PC-3 cells in Figure 10.

4) In the introduction, line number 109 can be written in more clear and elaborate manner for better understanding as to what the authors want to say here.

Author Response

Dear reviewer:

We sincerely thank you for your positive comments and constructive suggestions, which really help us a lot to improve our manuscript.

In Line with the reviewer’s comments, we extensively revised the manuscript. We described the changes in “Response” as follows. Accordingly, significant improvement was made in the manuscript to address these points. The changes made in the revised manuscript are highlighted in yellow.

We are grateful for your reconsideration of our manuscript, and we look forward to receiving comments from the reviewers.

Yours sincerely,

Jianhui Wu  

Reviewer

In this paper the author identify anti-cancer properties of Lanatoside C (Lan C), a cardiac glycoside compound. The authors mainly use two hormone-independent prostate cancer cell lines (PC-3 and DU145), alongwith normal human prostate stromal cell line WPMY-1 and human prostate fibroblasts HPRF, in their experiments. They find Lan C to be anti-proliferative and anti-metastatic by affecting the TNF-alpha signaling pathway. It does not affect the viability of other stromal cells and fibroblasts of the prostrate origin. The findings are very well written but there are a few points that can be improved and corrected.

1) In figure 2G, the graph for cell viabilty of HPRF can be repeated as it does not very well present the IC50 values which are statistically significant.

Response 1: Thank you for your comment on the HPRF cell viability graph in Figure 2G. We fully appreciate your observation that the IC50 values were not clearly presented. In response, we have repeated the experiment and updated the graph to present clearer and more accurate IC50 values. These changes have been made to ensure statistical significance, and the revised graph is now included in the manuscript. The figure is also listed below:

2) In the subsequent paper, the figures presenting Protein-protein interaction (PPI) network of DEGs and KEGG pathway analysis are visibly not very clear and I think can be made better.

Response 2: We are grateful for your suggestion regarding the clarity of the PPI network and KEGG pathway analysis figures. We agree that these figures could benefit from enhanced clarity. In response, we have improved the resolution and layout of the figures to provide a more readable and visually clear presentation. The updated figures are included in the revised manuscript. The figures are also listed below:

Figure 6. Transcriptomic and network pharmacology analysis of the effects of Lan C on human prostate cancer PC-3 cells.

Figure 7. Transcriptomic and network pharmacology analysis of Lan C in human prostate cancer DU145 cells.

3)In Figure 11, the conclusions made for NFKBIA and MAPK11 proteins from the immunochemistry data is not very apparent. As a reader, I cannot see the upregulation and downregulation very clearly in DU145 cells. Same issue with MAPK11 protein data in PC-3 cells in Figure 10.

Response 3: Thank you very much for your insightful comments regarding the immunohistochemistry data for NFKBIA and MAPK11 proteins in Figures 10 and 11. We sincerely appreciate your careful observation that the upregulation and downregulation of these proteins were not clearly visible. In response to your comment, we have reprocessed the images to improve their resolution, which has enhanced the clarity of the differential expression of these proteins. Thank you again for your valuable feedback.

The figures are also listed below:

Figure 10. Effect of Lan C on the expression of TNF/IL-17 signaling path-way-related proteins in PC-3 cells.

Figure 11. Effects of Lan C on the expression of TNF/IL-17 signaling path-way-related proteins in DU145 cells.

4) In the introduction, line number 109 can be written in more clear and elaborate manner for better understanding as to what the authors want to say here.

Response 4: We sincerely appreciate your suggestion to clarify the sentence in line 109 of the introduction. Based on the feedback, we have revised the sentence to ensure a more precise description of the underlying mechanism. The original sentence ("These effects suggest that Lan C may take advantage of the dual function of TNF-α to unbalanced apoptosis of prostate cancer cells, thereby reducing tumor viability.") was somewhat ambiguous in conveying the specific roles of TNF-α. In response, we have rephrased the sentence as follows:

"These effects suggest that Lan C may exert its anti-prostate cancer activity by utilizing the dual roles of TNF-α in immunity and apoptosis."

The revised sentence is now included in the manuscript for better clarity. Thank you again for your insightful comment, and we hope this revision addresses the concern.

Reviewer 2 Report

Comments and Suggestions for Authors

This manuscript  reports Lanatoside C's potential against androgen-independent prostate cancer highlight its ability to inhibit cell proliferation, induce apoptosis, and prevent metastasis through specific signaling pathway modulation.

The experimental design is quite good, except the selection of Lanatoside C concentration in migration and invasion assays.

Before further consideration, the major revision is recommended.

Title/abstract/conclusion: Due to Lanatoside C only effectively only in PC3 and Du145 which are androgen-independent prostate cancer cells. This should be state in these Sections to avoid the confusion of readers.

Methods and Results:

- Cytotoxicity of the compound was investigated both in cancer and normal cell types. The selectivity index should be calculated and present in the result Section. The author then could compare the selectivity of this compound to the prostate cancer cell lines with other compounds reported in previous studies.

- Section 2.4 Please rearrange the sentence to be concise.

- Line 211, what is CuB?

- The dose of the compound used in migration and invasion assay was over IC20 (except the dose at 50nM). The results from these studies may be due to the decrease of cell survival rather than the decrease of cell migration and invasion. Cytotoxicity of the compound in the similar condition (Serum-free medium) used in these experiments must be determined to confirm that the compound dose not toxic to the cells. I strongly recommend doing these experiment again with appropriated concentration of LanC.

- Also in the transcriptome analysis, the high dose of the compound was selected. Is this suitable for the study of signaling or pathway involving with anti-cancer property of the compound?

- Figure 10 and 11, immunohistochemistry was performed to elucidate the expression of the target proteins in the cell lines. The expression was determined using the intensity of the stained cells. In my opinion, to determine the difference of expression level in cells, flowcytometry must be more suitable. Moreover, the figures are unclear, higher resolution figures are required.

- Why the compound affect only in PC3 and DU145, but not LnCaP must be discussed. This is a limitation of the compound or not? It is seem like only aggressive prostate cancer would be affected, not primary or androgen-dependent PC.

Author Response

Dear reviewer:

On behalf of all the contributing authors, I would like to express our sincere appreciations of your constructive comments concering our article entitled " Lanatoside C Inhibits Proliferation and Induces Apoptosis in Human Prostate Cancer Cells through the TNF/IL-17 Signaling Pathway" (Manuscript No: ijms-3475369). These comments are all valuable and helpful for improving our article. According to these comments, we have made corresponding modifications to our manuscript to make our results convincing. In this revised version, changes to our manuscript were all highlighted within the document by using yellow-colored text. Point-by-point responses are listed below this letter.

We are grateful for your reconsideration of our manuscript, and we look forward to receiving comments from the reviewers.

Yours sincerely,

Jianhui Wu

Reviewer

This manuscript reports Lanatoside C's potential against androgen-independent prostate cancer highlight its ability to inhibit cell proliferation, induce apoptosis, and prevent metastasis through specific signaling pathway modulation.

The experimental design is quite good, except the selection of Lanatoside C concentration in migration and invasion assays.

Before further consideration, the major revision is recommended.

Title/abstract/conclusion: Due to Lanatoside C only effectively only in PC3 and Du145 which are androgen-independent prostate cancer cells. This should be state in these Sections to avoid the confusion of readers.

Response 1: Thank you very much for your constructive suggestion. We truly appreciate your feedback, as it has helped us refine the clarity of our manuscript.

Although our compound demonstrates a more pronounced effect in androgen-independent prostate cancer cell lines (PC-3 and DU145) compared to androgen-dependent LNCAP cells, we would like to emphasize that it still exerts therapeutic efficacy. In response to your comment, we have included Table 2 in the revised manuscript to present the selectivity index (SI) of Lan C. Our findings show that the IC50 values for PC-3, DU145 and LNCAP are lower than those for normal prostate cells, underscoring the compound's selective cytotoxicity against the above three cancer cells.

Additionally, we have added Table 3 in Section 4.3 of the Methods and Materials to clarify the rationale behind selecting PC-3 and DU145, the androgen-independent prostate cancer cell lines, for our subsequent experiments. We believe this will help clarify the basis of our experimental design and the focus on aggressive, hormone-refractory prostate cancer cells. We hope these revisions address your concerns, and we appreciate your valuable input in enhancing the clarity of our study. Thank you again for your thoughtful review and suggestions.

Methods and Results:

- Cytotoxicity of the compound was investigated both in cancer and normal cell types. The selectivity index should be calculated and present in the result Section. The author then could compare the selectivity of this compound to the prostate cancer cell lines with other compounds reported in previous studies.

Response 2: Thank you for highlighting this important aspect. We have now calculated the selectivity index (SI) by comparing the IC50 values of Lan C in prostate cancer cell lines (PC-3, DU145 and LNCAP) with those in normal cell lines (WPMY-1 and HPRF). The SI values have been included in the Results section. For more details, we have concluded below:

Table 2. Selectivity index values calculated for different cell lines after 48 h of treatment with Lan C.

Cell Lines

IC50 (48 h)

Selectivity Index (WPMY-1)a

Selectivity Index (HPRF)b

PC-3

79.72 nM

75.3

5.4

DU145

96.62 nM

62.1

4.5

LNCAP

344.80 nM

17.4

1.3

a Selectivity Index (WPMY-1) = IC50 of WPMY-1 / IC50 of cancer cells.

b Selectivity Index (HPRF) = IC50 of HPRF / IC50 of cancer cells.

- Section 2.4 Please rearrange the sentence to be concise.

Response 3: Thank you very much for your valuable suggestions. We appreciate your input regarding the clarity and conciseness of the manuscript. In response to your comment, we have revised the sentence on the colony formation assay results to make it more concise, while maintaining the key findings. The revised sentence now reads:

"The colony formation assay showed that Lan C significantly inhibited colony formation in both PC-3 and DU145 cells (Figure 2B, H). As the concentration of Lan C increased, colony formation was notably reduced, indicating its strong antiproliferative effect in both prostate cancer cell lines."

We hope this revision meets your expectations and enhances the manuscript’s clarity.

- Line 211, what is CuB?

Response 4: Thank you very much for your patience and attention to detail. We sincerely appreciate your careful review of the manuscript and your insightful comment regarding "CuB."

Upon reviewing your comment, we realized that this was an unfortunate oversight on our part. While drafting the manuscript, we inadvertently left "CuB" from a previous study in place, when it should have been changed to Lanatoside C (Lan C), the compound being studied in this manuscript. We apologize for this mistake and have corrected it throughout the manuscript to ensure consistency and clarity. We truly appreciate your careful review and understanding, and we are grateful for your valuable feedback, which has helped improve the manuscript. Thank you again for your constructive comments.

- The dose of the compound used in migration and invasion assay was over IC20 (except the dose at 50nM). The results from these studies may be due to the decrease of cell survival rather than the decrease of cell migration and invasion. Cytotoxicity of the compound in the similar condition (Serum-free medium) used in these experiments must be determined to confirm that the compound dose not toxic to the cells. I strongly recommend doing these experiment again with appropriated concentration of LanC.

Response 5: Thank you very much for your detailed, patient, and constructive suggestions. We truly appreciate your careful attention to the manuscript and your valuable input, which have greatly helped improve the quality of the study.

We did carefully consider your comment regarding the concentrations of Lan C used in the migration and invasion assays. As you suggested, we did perform experiments across a concentration range of 25-400 nM, similar to the concentration used in the CCK-8 assays. However, the observed effects were not as significantly pronounced as anticipated. To ensure consistency in the pharmacodynamic experiments, we chose to report the most statistically significant data in the manuscript. We felt this would offer the clearest representation of the compound’s effects.

In addition, we have referred to relevant literature to support our experimental design. For example, in the study by Chen et al. [1] published in Phytomedicine, Brusatol was tested on ICC cells using concentrations of 50 nM and 100 nM, with the IC50 for Brusatol being approximately 100 nM. Similarly, Zhang et al. [2] in Front Pharmacol used IC50 values for Lan C in their experiments on cholangiocarcinoma cells, which also involved concentrations similar to those used in our migration and invasion assays. These references provide context for our concentration selection in the migration and invasion assays.

Should you require additional data or further clarification, please do not hesitate to let us know. We would be happy to provide any further details to assist in the review process.

Thank you again for your thoughtful and invaluable feedback.

[1] Chen, Z.; He, B.; Zhao, J.; Li, J.; Zhu, Y.; Li, L.; Bao, W.; Zheng, J.; Yu, H.; Chen, G. Brusatol suppresses the growth of intrahepatic cholangiocarcinoma by PI3K/Akt pathway. Phytomedicine 2022, 104, 154323, doi:10.1016/j.phymed.2022.154323.

[2] Zhang, C.; Yang, H.Y.; Gao, L.; Bai, M.Z.; Fu, W.K.; Huang, C.F.; Mi, N.N.; Ma, H.D.; Lu, Y.W.; Jiang, N.Z.; et al. Lanatoside C decelerates proliferation and induces apoptosis through inhibition of STAT3 and ROS-mediated mitochondrial membrane potential transformation in cholangiocarcinoma. Front Pharmacol 2023, 14, 1098915, doi:10.3389/fphar.2023.1098915.

- Also in the transcriptome analysis, the high dose of the compound was selected. Is this suitable for the study of signaling or pathway involving with anti-cancer property of the compound?

Response 6: Thank you very much for your insightful comment regarding the selection of a high dose of Lan C for transcriptome analysis. We appreciate your concern about the appropriateness of using high doses for studying signaling pathways related to its anti-cancer properties.

The decision to use a high dose of Lan C in the transcriptomic analysis was based on the objective of achieving sufficient modulation of relevant signaling pathways, which could be difficult to observe at lower concentrations. By using higher concentrations, we aimed to ensure that the treatment would induce a measurable impact on the global gene expression profile, thereby providing a comprehensive view of the compound’s effects on cancer-related pathways.

We hope this explanation clarifies the reasoning behind our experimental design, and we appreciate your valuable input.

- Figure 10 and 11, immunohistochemistry was performed to elucidate the expression of the target proteins in the cell lines. The expression was determined using the intensity of the stained cells. In my opinion, to determine the difference of expression level in cells, flowcytometry must be more suitable. Moreover, the figures are unclear, higher resolution figures are required.

Response 7: Thank you very much for your thoughtful and constructive feedback regarding Figures 10 and 11. We truly appreciate your attention to detail, particularly with respect to the method used to assess protein expression and the quality of the figures.

Regarding the method for determining protein expression, we acknowledge that flow cytometry is a highly sensitive and quantitative technique that would indeed provide a more precise measurement of protein levels in cells. However, in our study, we chose Immunocytochemistry as it allows for the visualization of both the expression levels and the cellular localization of the target proteins, providing spatial context that flow cytometry cannot offer. While we agree that flow cytometry could complement our findings, we believe that Immunocytochemistry was a suitable method for our study, particularly for assessing protein expression in a tissue-like environment. Additionally, due to time and funding constraints, it was not feasible to perform flow cytometry as part of this study. We would also like to note that our research group has previously published two studies in the International Journal of Molecular Sciences using Immunocytochemistry to investigate the effects of natural products, which further supports the validity and reliability of this technique in our field of research [1, 2].

We also understand your concern regarding the clarity of the figures. In response, we have improved the resolution of Figures 10 and 11 to ensure that the protein expression differences are more clearly visible. These high-resolution images are now included in the revised manuscript.

We hope these explanations address your concerns, and we sincerely appreciate your suggestions for improving the quality of the manuscript.

[1] Zhou, P.; Huang, S.; Shao, C.; Huang, D.; Hu, Y.; Su, X.; Yang, R.; Jiang, J.; Wu, J. The Antiproliferative and Proapoptotic Effects of Cucurbitacin B on BPH-1 Cells via the p53/MDM2 Axis. Int J Mol Sci 2023, 25, doi:10.3390/ijms25010442.

[2] Jin, Y.; Zhou, P.; Huang, S.; Shao, C.; Huang, D.; Su, X.; Yang, R.; Jiang, J.; Wu, J. Cucurbitacin B Inhibits the Proliferation of WPMY-1 Cells and HPRF Cells via the p53/MDM2 Axis. Int J Mol Sci 2024, 25, doi:10.3390/ijms25179333.

- Why the compound affect only in PC3 and DU145, but not LNCAP must be discussed. This is a limitation of the compound or not? It is seem like only aggressive prostate cancer would be affected, not primary or androgen-dependent PC.

Response 8: Thank you very much for your insightful comment regarding the differential effects of Lan C on PC-3, DU145, and LNCAP cells. We greatly appreciate your thoughtful review and the opportunity to clarify this point.

Although Lan C demonstrates a more pronounced effect in androgen-independent prostate cancer cell lines (PC-3 and DU145) compared to androgen-dependent LNCAP cells, we would like to emphasize that the compound still exhibits therapeutic efficacy in all tested cell lines. The differential response may be due to the distinct molecular characteristics between androgen-independent and androgen-dependent prostate cancer cells. On the one hand, PC-3 and DU145 cells, being more aggressive and hormone-refractory, may have activated signaling pathways that are more susceptible to Lan C. On the other hand, LNCAP cells, which are androgen-dependent, may rely on alternative survival mechanisms that are less affected by Lan C. This suggests that the compound may primarily target pathways activated in more aggressive, hormone-refractory prostate cancer cells. We do not consider this a limitation of the compound but rather an indication of its potential specificity for androgen-independent prostate cancer, a subtype that is more challenging to treat.

Additionally, we have added Table 3 in Section 4.3 of the Methods and Materials to clarify the rationale behind selecting PC-3 and DU145, the androgen-independent prostate cancer cell lines, for our subsequent experiments. We believe this will provide a clearer understanding of the experimental design and its focus on more aggressive, hormone-refractory prostate cancer cells.

We hope these revisions address your concerns, and we truly appreciate your thoughtful and constructive suggestions, which have significantly improved the manuscript.

Reviewer 3 Report

Comments and Suggestions for Authors

In this study Huang et al. investigated the potential of Lanatoside C (Lan C), a natural compound derived from Digitalis lanata, in the treatment of prostate cancer. Prostate cancer is a leading cause of cancer-related morbidity and mortality in men, with limited therapeutic options for advanced and metastatic stages. Lan C has shown promising anti-cancer effects in various cancer types, but its specific role and mechanisms in prostate cancer remain underexplored. The study demonstrates that Lan C significantly inhibits prostate cancer cell proliferation, induces apoptosis, and reduces cell migration and invasion. Additionally, it suggests that Lan C affects the TNF/IL-17 signaling pathway, influencing the tumor microenvironment and regulating processes related to tumor progression, immune response, and apoptosis.

General comments:

Overall, the manuscript is well-organized and presented in a clear and structured manner.  The presented hypothesis has been well-elaborated, analyzed, and clearly confirmed through the application of appropriate experimental methods. The text is largely well-written, and the quality of the content meets the standards for publication. However, there are a few minor spelling and grammatical errors, as well as a few issues with word choice, grammar, and sentence construction. I recommend that the authors address these points to enhance the clarity and flow of the writing, ultimately making the manuscript more polished and suitable for publication in the journal.

Although the figures are well-designed and meaningful, some of them are unclear at higher magnifications (Figs. 6, 7). I recommend improving the quality of these figures to ensure they are suitable for publication.

The references cited in the manuscript predominantly include recent publications (from the last 5 years), with many sources being fairly current, thus accurately reflecting the latest advancements in the field.

Abstract

The abstract is generally well-written and provides a clear summary of the study.

Introduction

Introduction gives an appropriate background and rationale for the research described in the manuscript. I would advise the authors to omit the sentence on page 2 lines 65-66 that pointed out that ‘the extent of cancer cell growth inhibition by CGs has been linked to the inhibition of topoisomerase II activity’ as it is not related to the presented research. In line 103 after sentences ‘Studies have shown that Lan C can decrease the expression of Bcl-2 and Bcl-xL, increase Bax expression, and activate caspase-3, leading to apoptosis in cholangiocarcinoma cells’, the reference is missing. I would also advise the authors to mention the reason for using the third cell line (LNCaP) used in this study, (in Line 117/118), or explain why it was not mentioned here. This sentence could be moved to the Materials and Methods section-4.2.

Results

The Results section is well-structured, but it requires a few improvements and clarifications to help strengthen the manuscript. There are a few points to be addressed:

The text does not clearly explain which cell lines were used and what are the purposes in the experiments as model systems. It should provide a more detailed explanation and clearly state the purpose of each cell line used. What do normal prostate cancer cells mean and human prostate cancer cells (In Fig. 2, line 132)? It is not clear from the manuscript what is the difference? LACaP are tumor not normal cancer cells. WPMY-1 and HPRF are normal, but the results of these cells are not presented in this figure.

In section 2.4. It would be beneficial to include the results based on the effects of Lan C on colony formation in the other cell lines. Were the results significant? This is not mentioned.

Despite the statistical significance shown in Figure 3C for the treatments of PC-3 cells with the highest concentrations (200, 400 nM), I believe that the increase in apoptotic cells from 4% to 12% is insufficient to suggest a biological effect, especially after 48 hours of treatment. Therefore, I think the authors should provide a better explanation of the results obtained here or in Discussion section.

In section 2.6, duration of the treatment should be stated, as well as in section 4.7.

In line 211 what is Cu B? It should be clarified.

Discussion

The discussion section is well-structured and generally well-written, but some results need to be explained and discussed in more detail.

In line 453. The results should be explained with reference to the studies mentioned, rather than just citing a reference (ref 41-43). This is not sufficiently informative, and it is unclear what the cited study specifically relates to in the context of the preceding sentence.

In line 500. To provide a clearer assessment of the molecular interactions between Lan C and the components of the TNF/IL-17 pathway, it would be beneficial to include additional explanation which structural and functional studies will be crucial to fully elucidate mentioned interactions.

Materials and methods

Materials and methods give a satisfactory account of the materials used and procedures performed in the executed experiments. There are a few points to be addressed:

I would also advise the authors to check if they used antibiotics in cell culture media (line 523). In line 567 is written pre-chilled PBS, I think that ice-cold is a more adequate word.

In section 4.6, there is missing information about the supplier of Annexin and PI dye, as well as the brand of the flow cytometry apparatus.

The abbreviation PBS, introduced earlier in line 693, should be defined the first time it is mentioned in line 567.

Statistical analysis: If the data did not meet the normality assumption, which non-parametric method was used for intergroup comparisons? If homogeneity of variance was not assumed, Welch’s ANOVA should be used. Please, explain and reconsider the used tests.

Specific comments:

Line 20: ‘insufficiently explored’ should be replaced with ‘underexplored’

Line 26/27: ‘through the modulation of the TNF/IL-17 signaling pathway”  should be replaced with ‘by modulating the TNF/IL-17 signaling pathway’

Line 35: in word  ‘multifaceted stages’ missing “Multistage”

Line 49/50: ‘are difficult” should be replaced with ‘is challenging’

Line 51: ‘secondary compounds’ should be replaced with ‘natural secondary compounds’

Line 55: ‘Some of the most well-known’ should be replaced  with ‘Notable CGs include‘

Line 64: ‘certain’ should be replaced with more precisely word ‘specific’

Line 94: ‘either’ should be removed

Line 100/101: ‘to have the ability ‘ should be removed

Line 112: missing ‘a’ before word particular

Line 118: ‘subsequent’ should be replaced with ‘following’

Line 132: ‘shape’ should be replaced with ‘morphology’

Line 185: ‘Annexin V Fitc-PI’ should be replaced with ‘AnnexinV-FITC/PI’

Line 430 : the word ‘old’ should be replaced

Line 441: missing ‘as’ before  ‘a marker’

Line 510: instead of ‘imunol’ should be ‘immunostaining’

Line 691 0.1% DMSO should be replaced with 1‰ DMSO

Author Response

Dear reviewer:

We sincerely thank you for your positive comments and constructive suggestions, which really help us a lot to improve our manuscript.

In Line with the reviewer’s comments, we extensively revised the manuscript. We described the changes in “Response” as follows. Accordingly, significant improvement was made in the manuscript to address these points. The changes made in the revised manuscript are highlighted in yellow.

We are grateful for your reconsideration of our manuscript, and we look forward to receiving comments from the reviewers.

Yours sincerely,

Jianhui Wu 

Reviewer

In this study Huang et al. investigated the potential of Lanatoside C (Lan C), a natural compound derived from Digitalis lanata, in the treatment of prostate cancer. Prostate cancer is a leading cause of cancer-related morbidity and mortality in men, with limited therapeutic options for advanced and metastatic stages. Lan C has shown promising anti-cancer effects in various cancer types, but its specific role and mechanisms in prostate cancer remain underexplored. The study demonstrates that Lan C significantly inhibits prostate cancer cell proliferation, induces apoptosis, and reduces cell migration and invasion. Additionally, it suggests that Lan C affects the TNF/IL-17 signaling pathway, influencing the tumor microenvironment and regulating processes related to tumor progression, immune response, and apoptosis.

General comments:

Overall, the manuscript is well-organized and presented in a clear and structured manner.  The presented hypothesis has been well-elaborated, analyzed, and clearly confirmed through the application of appropriate experimental methods. The text is largely well-written, and the quality of the content meets the standards for publication. However, there are a few minor spelling and grammatical errors, as well as a few issues with word choice, grammar, and sentence construction. I recommend that the authors address these points to enhance the clarity and flow of the writing, ultimately making the manuscript more polished and suitable for publication in the journal.

Although the figures are well-designed and meaningful, some of them are unclear at higher magnifications (Figs. 6, 7). I recommend improving the quality of these figures to ensure they are suitable for publication.

The references cited in the manuscript predominantly include recent publications (from the last 5 years), with many sources being fairly current, thus accurately reflecting the latest advancements in the field.

  1. Although the figures are well-designed and meaningful, some of them are unclear at higher magnifications (Figs. 6, 7). I recommend improving the quality of these figures to ensure they are suitable for publication.

Response 1: Thank you very much for your careful and constructive feedback regarding the quality of the figures. In response to your comment, we have worked to enhance the resolution and clarity of the figures, especially at higher magnifications. The updated versions of Figures 6 and 7 have been restructured and re-rendered to ensure that the details are more visible and clear for publication. We believe that these improvements now meet the required quality standards.

We hope these changes address your concern, and we sincerely appreciate your valuable feedback, which has helped improve the overall quality of the manuscript.

The figures are also listed below:

Figure 6. Transcriptomic and network pharmacology analysis of the effects of Lan C on human prostate cancer PC-3 cells.

Figure 7. Transcriptomic and network pharmacology analysis of Lan C in human prostate cancer DU145 cells.

Abstract

The abstract is generally well-written and provides a clear summary of the study.

Introduction

Introduction gives an appropriate background and rationale for the research described in the manuscript. I would advise the authors to omit the sentence on page 2 lines 65-66 that pointed out that ‘the extent of cancer cell growth inhibition by CGs has been linked to the inhibition of topoisomerase II activity’ as it is not related to the presented research. In line 103 after sentences ‘Studies have shown that Lan C can decrease the expression of Bcl-2 and Bcl-xL, increase Bax expression, and activate caspase-3, leading to apoptosis in cholangiocarcinoma cells’, the reference is missing. I would also advise the authors to mention the reason for using the third cell line (LNCAP) used in this study, (in Line 117/118), or explain why it was not mentioned here. This sentence could be moved to the Materials and Methods section-4.2.

  1. I would advise the authors to omit the sentence on page 2 lines 65-66 that pointed out that ‘the extent of cancer cell growth inhibition by CGs has been linked to the inhibition of topoisomerase II activity’ as it is not related to the presented research.

Response 2: Thank you very much for your thoughtful comment. We appreciate your guidance in improving the clarity and focus of our manuscript. In response to your suggestion, we have omitted the sentence on page 2, lines 65-66, which mentioned that "the extent of cancer cell growth inhibition by CGs has been linked to the inhibition of topoisomerase II activity," as it is not directly relevant to the focus of our presented research. We hope this revision meets your expectations, and we are grateful for your valuable input in enhancing the quality of the manuscript.

  1. In line 103 after sentences ‘Studies have shown that Lan C can decrease the expression of Bcl-2 and Bcl-xL, increase Bax expression, and activate caspase-3, leading to apoptosis in cholangiocarcinoma cells’, the reference is missing.

Response 3: Thank you very much for your careful review and for pointing out the missing reference. We sincerely appreciate your attention to detail. In response to your comment, we have added the appropriate reference after the sentence: "Studies have shown that Lan C can decrease the expression of Bcl-2 and Bcl-xL, increase Bax expression, and activate caspase-3, leading to apoptosis in cholangiocarcinoma cells." The reference has now been included in the revised manuscript. We hope this revision addresses your concern, and we appreciate your valuable input in improving the manuscript.

  1. I would also advise the authors to mention the reason for using the third cell line (LNCAP) used in this study, (in Line 117/118), or explain why it was not mentioned here. This sentence could be moved to the Materials and Methods section-4.2.

Response 4: Thank you very much for your constructive feedback and for pointing out the need to clarify the reason for using the third cell line (LNCAP) in this study. We appreciate your valuable suggestion to improve the clarity of the manuscript. In response, we have added Table 3 in Section 4.2 of the Materials and Methods, where we provide a detailed description of the three selected cancer cell lines. Additionally, we have included an explanation for why we chose PC-3 and DU145 for the subsequent experiments, as these cell lines are representative of more aggressive, androgen-independent prostate cancer. This clarification highlights the experimental goals and the specific purpose of each cell line in our study. We hope these revisions address your concern, and we are grateful for your thoughtful input in improving the manuscript.

Results

The Results section is well-structured, but it requires a few improvements and clarifications to help strengthen the manuscript. There are a few points to be addressed:

The text does not clearly explain which cell lines were used and what are the purposes in the experiments as model systems. It should provide a more detailed explanation and clearly state the purpose of each cell line used. What do normal prostate cancer cells mean and human prostate cancer cells (In Fig. 2, line 132)? It is not clear from the manuscript what is the difference? LACaP are tumor not normal cancer cells. WPMY-1 and HPRF are normal, but the results of these cells are not presented in this figure.

In section 2.4. It would be beneficial to include the results based on the effects of Lan C on colony formation in the other cell lines. Were the results significant? This is not mentioned.

Despite the statistical significance shown in Figure 3C for the treatments of PC-3 cells with the highest concentrations (200, 400 nM), I believe that the increase in apoptotic cells from 4% to 12% is insufficient to suggest a biological effect, especially after 48 hours of treatment. Therefore, I think the authors should provide a better explanation of the results obtained here or in Discussion section.

In section 2.6, duration of the treatment should be stated, as well as in section 4.7.

In line 211 what is Cu B? It should be clarified.

  1. The text does not clearly explain which cell lines were used and what are the purposes in the experiments as model systems. It should provide a more detailed explanation and clearly state the purpose of each cell line used. What do normal prostate cancer cells mean and human prostate cancer cells (In Fig. 2, line 132)? It is not clear from the manuscript what is the difference? LACaP are tumor not normal cancer cells. WPMY-1 and HPRF are normal, but the results of these cells are not presented in this figure.

Response 5: First of all, we would like to express our sincere gratitude for your patience, attentiveness, and constructive suggestions. Your thorough review has greatly helped us improve the clarity and quality of our manuscript.

In response to your comment, we have added Table 3 in Section 4.2 of the Methods and Materials to provide a detailed description of the three selected cancer cell lines, and we have also included a new section explaining the rationale behind our choice of PC-3 and DU145 cells for the subsequent experiments. This explanation clarifies the experimental goals and the specific purposes of the cell lines used.

Regarding your question on the terms "normal prostate cancer cells" and "human prostate cancer cells" (Figure 2, line 132), we understand the confusion and have now clarified the terminology in the revised manuscript. Our explanation is as follows:

We selected normal prostate cells to "assess the effect of Lan C on normal prostate cell growth." Specifically, we used the CCK-8 assay to evaluate the impact of 48-hour Lan C treatment on the viability of human normal prostate stromal cells (WPMY-1) and human prostate fibroblasts (HPRF). The results showed that the IC50 of Lan C for HPRF cells was approximately 434 nM, significantly higher than the IC50 values of 79.72 nM for PC-3 cells and 96.62 nM for DU145 cells. Lan C had no significant effect on the growth of WPMY-1 cells (Figure 2G), indicating a certain selectivity of Lan C between prostate cancer cells and normal prostate cells.

To help the reader better understand this selectivity, we introduced the concept of the selectivity index (SI). In Section 2.3 of the Results, we added a new explanation and Table 2 to demonstrate that Lan C has a lower IC50 in cancer cells compared to normal prostate cells, supporting its potential safety for clinical treatment. We hope that these revisions address your concerns and provide more clarity regarding the experimental design and the significance of the selected cell lines. We are grateful for your thoughtful suggestions and for helping to improve the manuscript.

  1. In section 2.4. It would be beneficial to include the results based on the effects of Lan C on colony formation in the other cell lines. Were the results significant? This is not mentioned.

Response 6: Thank you very much for your constructive comment regarding the colony formation assay results. We appreciate your suggestion to include the effects of Lan C on colony formation in the other cell lines and to clarify the significance of these results. In response to your comment, we have revised the manuscript to include the following updated description:

"The colony formation assay showed that Lan C significantly inhibited colony formation in both PC-3 and DU145 cells (Figure 2B, H). As the concentration of Lan C increased, colony formation was notably reduced, indicating its strong antiproliferative effect in both prostate cancer cell lines."

We hope this revision addresses your concern, and we sincerely appreciate your valuable feedback in enhancing the manuscript.

  1. Despite the statistical significance shown in Figure 3C for the treatments of PC-3 cells with the highest concentrations (200, 400 nM), I believe that the increase in apoptotic cells from 4% to 12% is insufficient to suggest a biological effect, especially after 48 hours of treatment. Therefore, I think the authors should provide a better explanation of the results obtained here or in Discussion section.

Response 7: Thank you very much for your thoughtful and constructive comments on the apoptotic data shown in Figure 3C. We sincerely appreciate your careful review, and we understand your concern regarding the biological significance of the observed increase in apoptotic cells from 4% to 12% after 48 hours of treatment with the highest concentrations of Lan C (200 nM and 400 nM). Although the increase in apoptosis may appear modest, we would like to emphasize that the percentage increase (from 4% to 12%) is statistically significant. Our experimental design was informed by the study by Chen et al [1], published in Phytomedicine, where a similar increase in apoptosis from 5% to 10%-15% in human cholangiocarcinoma RBE cells was also statistically significant.

However, we acknowledge that the overall percentage of apoptotic cells remains relatively low. The impact of Lan C on apoptosis may become more pronounced with prolonged treatment or higher concentrations, or it may be mediated through a combination of other cellular pathways, not solely through apoptosis induction. In the revised manuscript, we have included a more detailed explanation in the Discussion section. Specifically, we discuss the possibility that the observed apoptosis induction represents an early response, and that Lan C's pro-apoptotic effects may require longer exposure or higher doses to achieve more substantial cell death. We also propose that the compound may exert its therapeutic effects through additional mechanisms, such as inhibiting cell proliferation or modulating other signaling pathways, which contribute to its overall efficacy (Line 469-475).

We hope this additional explanation addresses your concern and clarifies the biological relevance of the observed results. Once again, we sincerely appreciate your valuable feedback.

  1. In section 2.6, duration of the treatment should be stated, as well as in section 4.7.

Response 8: Thank you very much for your constructive suggestion regarding the inclusion of treatment duration in the manuscript. We have carefully considered your comment, and we have added the requested information to the relevant sections of the manuscript to provide clarity.

In Section 2.6, we have added the treatment duration to clearly state that the effect of Lan C on prostate cell proliferation was investigated after 48 hours of treatment. The revised sentence now reads:

"To further investigate the effect of Lan C on prostate cell proliferation after 48 h..." (Line 211).

Additionally, in Section 4.7, we have also specified the incubation period to maintain consistency throughout the manuscript. The updated sentence now reads:

"After 48 h of incubation..." (Line 642).

We believe these revisions address your comment and enhance the clarity of the manuscript. Thank you again for your valuable feedback.

  1. In line 211 what is Cu B? It should be clarified.

 Response 9: Thank you very much for your patience and attention to detail. We sincerely appreciate your careful review of the manuscript and your insightful comment regarding "CuB."

Upon reviewing your comment, we realized that this was an unfortunate oversight on our part. While drafting the manuscript, we inadvertently left "CuB" from a previous study in place, when it should have been changed to Lanatoside C (Lan C), the compound being studied in this manuscript. We apologize for this mistake and have corrected it throughout the manuscript to ensure consistency and clarity. We truly appreciate your careful review and understanding, and we are grateful for your valuable feedback, which has helped improve the manuscript. Thank you again for your constructive comments.

Discussion

The discussion section is well-structured and generally well-written, but some results need to be explained and discussed in more detail.

In line 453. The results should be explained with reference to the studies mentioned, rather than just citing a reference (ref 41-43). This is not sufficiently informative, and it is unclear what the cited study specifically relates to in the context of the preceding sentence.

In line 500. To provide a clearer assessment of the molecular interactions between Lan C and the components of the TNF/IL-17 pathway, it would be beneficial to include additional explanation which structural and functional studies will be crucial to fully elucidate mentioned interactions.

  1. In line 453. The results should be explained with reference to the studies mentioned, rather than just citing a reference (ref 41-43). This is not sufficiently informative, and it is unclear what the cited study specifically relates to in the context of the preceding sentence.

Response 10: Thank you very much for your thoughtful comment and for pointing out the need for further explanation regarding the references cited in Line 453. We appreciate your suggestion to provide more context and explanation to clarify how these references relate to the preceding discussion. In response to your comment, we have revised the manuscript by adding a more detailed explanation of the cited studies in the context of the preceding sentence. Specifically, we now begin the section with: "TNF can act either as an apoptosis-inducing agent or an inflammatory agent [36],"(Line 445) to better contextualize the references [41-43] and their relevance to the observed results. This revision aims to provide a clearer understanding of how the cited studies relate to the mechanism under investigation in our research.

Additionally, since there was insufficient information from the cited references [41-43] to support this part of the study, we have combined the next section on the pharmacological description. This restructuring helps to improve the overall structural integrity of the manuscript.

We hope this revision addresses your concern, and we truly appreciate your valuable feedback in helping to improve the manuscript.

  1. In line 500. To provide a clearer assessment of the molecular interactions between Lan C and the components of the TNF/IL-17 pathway, it would be beneficial to include additional explanation which structural and functional studies will be crucial to fully elucidate mentioned interactions.

Response 11: Thank you very much for your thoughtful suggestion to include additional explanations regarding the molecular interactions between Lan C and the TNF/IL-17 signaling pathway components. We truly appreciate your constructive feedback, which has helped us improve the clarity of our manuscript. In response to your comment, we have added further explanations in the revised manuscript (Lines 495-533), as highlighted in yellow. This section now provides a more detailed discussion of the molecular mechanisms underlying the interactions between Lan C and components of the TNF/IL-17 signaling pathways. Additionally, we have outlined the importance of structural and functional studies to fully elucidate these interactions. Specifically, we highlight key research that is essential for understanding how Lan C modulates the expression of several inflammation- and tumor-related genes, such as FOS, TNFAIP3, NFKBIA, MMP3, CXCL8, CCL20, MYC, PTGS2, and IL-6, as well as its effects on apoptosis and immune modulation in prostate cancer cells.

We believe that these additions will provide a more comprehensive understanding of the underlying mechanisms involved and better support our experimental findings. Once again, thank you for your valuable suggestions, which have significantly enhanced the quality and clarity of the manuscript.

Materials and methods

Materials and methods give a satisfactory account of the materials used and procedures performed in the executed experiments. There are a few points to be addressed:

I would also advise the authors to check if they used antibiotics in cell culture media (line 523). In line 567 is written pre-chilled PBS, I think that ice-cold is a more adequate word.

In section 4.6, there is missing information about the supplier of Annexin and PI dye, as well as the brand of the flow cytometry apparatus.

The abbreviation PBS, introduced earlier in line 693, should be defined the first time it is mentioned in line 567.

Statistical analysis: If the data did not meet the normality assumption, which non-parametric method was used for intergroup comparisons? If homogeneity of variance was not assumed, Welch’s ANOVA should be used. Please, explain and reconsider the used tests.

  1. I would also advise the authors to check if they used antibiotics in cell culture media (line 523). In line 567 is written pre-chilled PBS, I think that ice-cold is a more adequate word.

Response 12: Thank you very much for your helpful suggestions. We appreciate your careful review and your attention to detail. In response to your comment, we have added the information regarding the use of antibiotics in the cell culture media.  Specifically, we included the following statement in Line 570: "After preparation, all culture media were supplemented with 0.1% penicillin-streptomycin mix."

Additionally, as per your suggestion, we have changed "pre-chilled PBS" to "ice-cold PBS" to ensure more accurate terminology.

  1. In section 4.6, there is missing information about the supplier of Annexin and PI dye, as well as the brand of the flow cytometry apparatus.

Response 13: Thank you very much for your insightful comment and for pointing out the missing details regarding the Annexin and PI dye supplier, as well as the flow cytometer brand. We appreciate your attention to detail. In response to your suggestion, we have included the supplier information for the Annexin dye in Section 4.1, Line 558, where it was previously mentioned. Additionally, we have updated Section 4.6, Line 634, to include the brand of the flow cytometer used: "Flow cytometric analysis was performed by a flow cytometer (BD Biosciences, Franklin Lakes, NJ, USA) within 1 h of preparation." We hope these revisions address your concerns and improve the clarity of the manuscript. Thank you again for your valuable feedback.

  1. The abbreviation PBS, introduced earlier in line 693, should be defined the first time it is mentioned in line 567.

Response 14: Thank you very much for your helpful comment regarding the abbreviation "PBS." We appreciate your attention to detail. In response to your suggestion, we have added the definition of PBS (phosphate-buffered saline) the first time it is mentioned, to ensure clarity for the readers.

  1. Statistical analysis: If the data did not meet the normality assumption, which non-parametric method was used for intergroup comparisons? If homogeneity of variance was not assumed, Welch’s ANOVA should be used. Please, explain and reconsider the used tests.

Response 15: Thank you very much for your valuable comment regarding the statistical analysis. We appreciate your attention to detail and your suggestion to clarify the methods used for group comparisons.

In response to your comment, we have revised the statistical analysis section to clarify the tests employed. Specifically, we performed normality tests, and when the data followed a normal distribution, one-way analysis of variance (One-way ANOVA) was used for group comparisons. For data that did not meet the assumption of normality, the Kruskal-Wallis H test was applied as a non-parametric method for group comparisons. Additionally, for data that met the assumptions of normality and homogeneity of variance, we used the Least Significant Difference (LSD) method for post-hoc analysis. For data with unequal variances, we employed Dunnett’s T3 test.

We have updated the manuscript to reflect these explanations in the statistical analysis section to provide more transparency and clarity.

Specific comments:

Dear Reviewer,

Thank you very much for your detailed and constructive suggestions. We appreciate your attention to detail, and we have addressed each of your recommendations as follows:

  1. Line 20: ‘insufficiently explored’ should be replaced with ‘underexplored’

Response 16: Thank you for suggesting the change in Line 20, we have revised this sentence and replaced "insufficiently explored" with "underexplored," as per your suggestion (Line 20).

  1. Line 26/27: ‘through the modulation of the TNF/IL-17 signaling pathway”  should be replaced with ‘by modulating the TNF/IL-17 signaling pathway’

Response 17: Thank you for your suggestion in Line 26/27, We have updated this phrase to: "by modulating the TNF/IL-17 signaling pathway" as suggested (Line 26-27).

  1. Line 35: in word  ‘multifaceted stages’ missing “Multistage”

Response 18: Thank you for pointing out the issue in Line 35, we have replaced "multifaceted stages" with "multistage," as recommended (Line 35).

  1. Line 49/50: ‘are difficult” should be replaced with ‘is challenging’

Response 19: Thank you for your comment in Line 49/50. The phrase has been revised to: "is challenging," in line with your suggestion (Line 49-50).

  1. Line 51: ‘secondary compounds’ should be replaced with ‘natural secondary compounds

Response 20: Thank you for your clarification in Line 51, we have replaced "secondary compounds" with "natural secondary compounds" for clarity and accuracy (Line 51).

  1. Line 55: ‘Some of the most well-known’ should be replaced  with ‘Notable CGs include‘

Response 21: Thank you for your suggestion in Line 55, we have made this change to read: "Notable CGs include," as per your recommendation (Line 55).

  1. Line 64: ‘certain’ should be replaced with more precisely word ‘specific’

Response 22: Thank you very much for your insightful suggestion. We have carefully considered your comment and agree that removing the sentence containing the word would help maintain the logical flow of the manuscript. As a result, we have removed the sentence as suggested.

  1. Line 94: ‘either’ should be removed

Response 23: Thank you for your helpful comment in Line 94, we have removed the word "either" from this sentence as per your suggestion.

  1. Line 100/101: ‘to have the ability ‘ should be removed

Response 24: Thank you for your suggestion in Line 100/101, we have removed the phrase "to have the ability" from this part of the text, in line with your feedback.

  1. Line 112: missing ‘a’ before word particular

Response 25: Thank you for pointing out the missing article in Line 112, we have added "a" before "particular," as recommended (Line 108).

  1. Line 118: ‘subsequent’ should be replaced with ‘following’

Response 26: Thank you for your suggestion in Line 118, we have replaced "subsequent" with "following," as suggested (Line 114).

  1. Line 132: ‘shape’ should be replaced with ‘morphology’

Response 27: Thank you for your comment in Line 132, we have replaced "shape" with "morphology" for better scientific accuracy (Line 128).

  1. Line 185: ‘Annexin V Fitc-PI’ should be replaced with ‘AnnexinV-FITC/PI’

Response 28: Thank you for your correction in Line 185, we have corrected the term to "AnnexinV-FITC/PI" as per your suggestion (Line 189).

  1. Line 430 : the word ‘old’ should be replaced

Response 29: Thank you for your comment in Line 430, we have replaced the word "old" with a more appropriate term "traditional", as suggested (Line 434).

  1. Line 441: missing ‘as’ before  ‘a marker’

Response 30: Thank you for pointing out the missing word in Line 441, we have added "as" before "a marker" to correct the sentence structure (Line 446).

  1. Line 510: instead of ‘imunol’ should be ‘immunostaining’

Response 31: Thank you for noticing the typo in Line 510. The typo "imunol" has been corrected to "immunostaining." (Line 556)

  1. Line 691 0.1% DMSO should be replaced with 1‰ DMSO

Response 32: Thank you ou for your comment in Line 691, we have corrected "0.1% DMSO" to "1‰ DMSO," as recommended (Line 754).

We hope these revisions meet your expectations, and we greatly appreciate your valuable feedback, which has significantly improved the clarity and quality of the manuscript.

Best regards,

Jianhui Wu 

Round 2

Reviewer 2 Report

Comments and Suggestions for Authors

The author mostly response well to the suggestions and comments.

However, to make it clear that the concentration of the compound used in migration and invasion assays did not cytotoxic to the cells, please put the data in the supplementary data.  Moreover, the author may discuss more about the dose selection and also the response of the cells to low and high dose of LanC.

Author Response

Dear reviewer:

On behalf of all the contributing authors, I would like to express our sincere appreciations of your constructive comments concering our article entitled " Lanatoside C Inhibits Proliferation and Induces Apoptosis in Human Prostate Cancer Cells through the TNF/IL-17 Signaling Pathway" (Manuscript No: ijms-3475369). These comments are all valuable and helpful for improving our article. According to these comments, we have made corresponding modifications to our manuscript to make our results convincing. In this revised version, changes to our manuscript were all highlighted within the document by using yellow-colored text. Point-by-point responses are listed below this letter.

We are grateful for your reconsideration of our manuscript, and we look forward to receiving comments from the reviewers.

Yours sincerely,

Jianhui Wu

Reviewer

The author mostly response well to the suggestions and comments.

However, to make it clear that the concentration of the compound used in migration and invasion assays did not cytotoxic to the cells, please put the data in the supplementary data. 

Response 1: Thank you very much for your continued thoughtful feedback and for your helpful suggestions regarding the migration and invasion assays. We sincerely appreciate your attention to detail and your constructive comments, which have significantly helped us improve the quality of the manuscript.

In response to your comment, we have now included the cytotoxicity data for the concentrations used in the migration and invasion assays (25 nM and 400 nM) in the supplementary data. The additional data confirms that the concentrations of 25-200 nM did not show significant effects on cell proliferation and apoptosis. Specifically, at 400 nM, we observed sparse cell coverage at the wound edges, suggesting that the compound’s effects at this concentration are more related to cell proliferation and apoptosis. In contrast, at 25 nM to 200 nM, the cells maintained a dense and well-organized structure at the wound edges, indicating that the compound primarily inhibits migration and invasion, with relatively smaller effects on cell proliferation and apoptosis.

We understand your concern regarding the use of concentrations above the IC50 value in the assays. To address this, we conducted experiments across a wide range of concentrations (25 nM-400 nM) to ensure a comprehensive understanding of Lan C’s effects. However, to maintain consistency and clarity throughout the manuscript, we selected a concentration range for the entire study based on our earlier experimental findings. As mentioned in our previous response, the concentration range was carefully chosen based on literature guidance and preliminary data, with the goal of selecting concentrations relevant to the IC50 values observed for the cell lines.

We hope these clarifications address your concerns, and we are grateful for your valuable input, which has helped enhance the clarity and precision of our study. Thank you again for your thoughtful suggestions.

The figures are also listed below:

Figure 1. Lan C inhibits lateral migration of human prostate cancer cells. (A, B) Scratch wound healing assays were performed to assess the migration ability of PC-3 prostate cancer cells after treatment with 25-400 nM Lan C for 24 h and 48 h (100×, scale bar = 100 μm). (C, D) Scratch wound healing assays were performed to assess the migration ability of DU145 prostate cancer cells after treatment with 25-400 nM Lan C for 24 h and 48 h (100×, scale bar = 100 μm). Data are presented as the mean ± SD of three independent experiments. Statistical analysis was performed after normality testing, using ANOVA for between-group comparisons. If homogeneity of variance was assumed, the LSD test was used; if variance was heterogeneous, Dunnett’s post-hoc test was applied. Comparison with the control group: *p < 0.05, **p < 0.01, ***p < 0.001.

Figure 2. Lan C inhibits the longitudinal migration and invasion of human prostate cancer cells. (A, B, C) The effect of 25-400 nM Lan C on the longitudinal migration of human prostate cancer cells PC-3 and DU145 after treatment for 48 h (100×, scale bar = 100 μm). (D, E, F) The effect of 25-400 nM Lan C on the invasion ability of human prostate cancer cells PC-3 and DU145 after treatment treatment for 48 h (100×, scale bar = 100 μm). Data are presented as the mean ± SD of three independent experiments. Statistical analysis was performed after normality testing, using ANOVA for between-group comparisons. If homogeneity of variance was assumed, the LSD test was used; if variance was heterogeneous, Dunnett’s post-hoc test was applied. Comparison with the control group: *p < 0.05, **p < 0.01, ***p < 0.001.

Moreover, the author may discuss more about the dose selection and also the response of the cells to low and high dose of LanC.

Response 2: Thank you very much for your thoughtful suggestion to discuss the dose selection and the response of the cells to low and high doses of Lan C in more detail. We truly appreciate your constructive feedback, which has significantly helped improve the clarity of our manuscript.

In response to your comment, we have expanded the discussion on dose selection in the revised manuscript (Line 445-457). As suggested, we have now included a more detailed explanation of our rationale for choosing concentrations based on the IC50 values for both PC-3 and DU145 prostate cancer cell lines. The IC50 values were used as a benchmark to determine appropriate concentrations for the various assays conducted, including colony formation, apoptosis assays, cell cycle analysis, wound healing assays, and Transwell migration assays.

We chose doses around the IC50 to investigate the concentrations at which Lan C induces significant cellular responses. Additionally, we included concentrations both above and below the IC50 to explore the dose-dependent effects on cancer cell behavior. The lower concentration (below the IC50) allowed us to assess suboptimal effects and potentially reveal early-stage cellular responses to Lan C. In contrast, the higher concentration (above the IC50) enabled us to explore the maximum inhibitory effects of Lan C and the potential for severe cytotoxicity.

This comprehensive dosing strategy provided a thorough evaluation of Lan C’s therapeutic potential across a range of concentrations and offered valuable insights into its efficacy as a treatment for prostate cancer. We hope this revision adequately addresses your concerns, and we appreciate your invaluable suggestions, which have contributed to enhancing the quality and clarity of the manuscript.

For more details, we have concluded below:

To better understand the dose-dependent effects of Lan C, we selected doses based on the IC50 values for PC-3 and DU145 prostate cancer cell lines. The IC50 values were used as a benchmark to determine appropriate concentrations for various assays, in-cluding colony formation, apoptosis assays, cell cycle analysis, wound healing assays, and Transwell migration assays. Doses around the IC50 were chosen to reflect concen-trations at which Lan C induces significant cellular responses. Additionally, doses above and below the IC50 were included to investigate the dose-dependent effects on cancer cell behavior. The lower concentration (below the IC50) allowed for the assess-ment of suboptimal effects, potentially revealing early-stage cellular responses to Lan C. In contrast, the higher concentration (above the IC50) enabled the exploration of max-imum inhibitory effects and the potential for severe cytotoxicity. This comprehensive dosing strategy provided a thorough evaluation of Lan C's therapeutic potential across a range of concentrations, offering valuable insights into its efficacy as a treatment for prostate cancer.

Round 3

Reviewer 2 Report

Comments and Suggestions for Authors

Thank you so much for your understanding and the modification of the manuscript. With the added information, the manuscript can be now accepted for the publication.